# Occurrence of pressure-forced meteotsunami events in the eastern Yellow Sea during 2010–2019

Myung-Seok Kim[1], Seung-Buhm Woo[1], Hyunmin Eom[2], and Sung Hyup You[2]

[1]Department of Ocean Sciences, Inha University, Incheon, Republic of Korea
[2]Marine Meteorology Division, Korea Meteorological Administration, Seoul, Republic of Korea

*Correspondence to*: Seung-Buhm Woo (sbwoo@inha.ac.kr)

**Abstract.** This study examined the occurrence of meteotsunamis in the eastern Yellow Sea and the conceptual framework of a monitoring/warning system. Using 1 min intervals of mean sea level pressure and sea level observations from 89 meteorological stations and 16 tide gauges between 2010 and 2019, a total of 42 pressure-forced meteotsunami events were classified. Most meteotsunamis (71%) displayed a distinct seasonal pattern occurring from March to June and intense meteotsunamis typically occurred at harbor tide gauges. The occurrence characteristics of the meteotsunamis were examined to improve the meteotsunami monitoring/warning system. Air pressure disturbances with speeds of 11–26 m/s and NNW–SW directions were conducive to meteotsunami generation. Most meteotsunamis (88%), as well as strong meteotsunamis with a wave height exceeding 40 cm (19%), had dominant period bands of less than 30 min, containing the resonant periods of harbors in the eastern Yellow Sea. Thus, the eastern Yellow Sea is a "harbor meteotsunami" dominated environment, characterized by frequent meteotsunami occurrences and local amplification in multiple harbors. This study can provide practical guidance on operation periods, potential hot spots, and risk levels to monitoring/warning system operators in the eastern Yellow Sea.

## 1 Introduction

Globally, monitoring high-frequency sea-level oscillations is crucial for warning system operators and policymakers (Šepić et al., 2015b) as floods occur frequently in coastal communities (Vilibić et al., 2014). High-frequency sea-level oscillations, such as infragravity waves, seiches, tsunamis, and meteotsunamis, have periods of several minutes to several hours (Rabinovich, 2009). Among them, meteotsunamis are high-frequency and tsunami-like sea-level oscillations (Monserrat et al., 2006) that are dominant in the tsunami frequency band (2 min to 2 h). However, unlike tsunami waves of seismic origin, meteotsunamis are atmospherically generated and amplified by multi-resonant mechanisms (Pattiaratchi and Wijeratne, 2015). Meteotsunamis occur by a well-known, three-stage mechanism (Monserrat et al., 2006). Initially, long waves are generated by air pressure disturbances in the open sea. Subsequently, these propagating long ocean waves are locked to the air pressure disturbance with a similar speed, causing resonance amplification, specifically the Proudman resonance (Proudman, 1929). Finally, internal resonance occurs between the dominant period of the pre-amplified waves and the fundamental periods of shelves, bays, or harbors. As a result, sea-level oscillations of several centimeters in the open sea can be increased to destructive amplitudes of

several meters along the shoreline. Pressure-forced meteotsunamis occur more frequently, both temporally and spatially, than seismic tsunamis (Pattiaratchi and Wijeratne, 2015) and have been consistently reported (Vilibić et al., 2021).

In the eastern Yellow Sea, destructive meteotsunami events on March 31, 2007, and May 4, 2008, caused severe loss of human life and property damage (Eom et al., 2012). On March 31, 2007, maximum wave heights of 1–3 m were detected at most tide gauges from midnight to dawn (Choi et al., 2008). Concurrently, strong air pressure disturbances (rate of pressure
change of 1.7–4.8 hPa/10 min) with a similar spatial scale propagated to multiple meteorological stations (Kim et al., 2019). It is currently the strongest meteotsunami event ever reported in the Yellow Sea; however, if there had been any meteotsunami monitoring system, the damage could have been reduced. The following year, at noon on Sunday, May 4, 24 people fishing near a breakwater were swept away by sudden meteotsunamis, causing 9 deaths and 15 injuries. This event demonstrated the importance of timing of meteotsunami occurrence because they are more dangerous when coastal areas are crowded with
people, especially on weekends. The sea-level oscillations observed at the tide gauges near the accident site were not significantly amplified (Choi and Lee, 2009), but the maximum wave height at the accident site, measured from coastal CCTV camera videos, was approximately 1.3 m (Yoo et al., 2010). Although relatively weak air pressure disturbances (1.8–1.9 hPa/10 min) on a local spatial scale were found to be related to the meteotsunamis (Kim et al., 2019), the mechanism for generating such amplified waves near the coast was unknown at that time. The potential coastal hazards caused by meteotsunamis became
known to the Korean public after the damage captured by the CCTV cameras was released to the media. To date, several previous studies in the eastern Yellow Sea have focused on determining the causes of events in which human and property damage were reported. However, meteotsunamis that do not cause notable damage can occur.

Understanding the temporal and spatial trends in meteotsunami occurrence is essential for the prevention of and preparation for potential coastal hazards (Linares et al., 2016). Accordingly, there have been attempts to develop a monitoring
system for meteotsunami disaster prevention by finding favorable atmospheric conditions that can cause potential meteotsunamis at various times and locations. This is because meteotsunamis are related to air-sea interactions, especially in the 1st and 2nd stages of the mechanism mentioned above. For example, a monitoring possibility was suggested based on correlations between synoptic atmospheric patterns and wave heights observed in the strongest meteotsunami events in the Balearic Islands, the Mediterranean, and the English Channel (Jansà et al., 2007; Šepić et al., 2012; Ozsoy et al., 2016; Vilibić
et al., 2018). A more realistic and quantitative approach, from the perspective of real-time assessment, used the existing meteorological stations in the Adriatic Sea to demonstrate that appropriate warnings can be issued by relating the characteristics of air pressure disturbances (e.g., intensity, speed, incoming direction) to the five levels of a meteotsunami danger (Šepić and Vilibić, 2011). Bechle et al. (2015) suggested common storm structures favorable to meteotsunami occurrence in Lake Michigan by using meteorological station data as well as the temporal and spatial patterns of reflectivity in radar images. In
addition, numerical model runs have been conducted to assess the vulnerability and risks in coastal areas for various propagation scenarios for atmospheric disturbances (Linares et al., 2016; Šepić et al., 2015a; Vilibić et al., 2005). Recently, an extreme sea-level hazard assessment was suggested based on deterministic atmospheric and ocean models as well as a statistical model, as providing a new avenue for meteotsunami early warning systems (Denamiel et al., 2019).

For meteotsunami disaster prevention in the eastern Yellow Sea, a real-time air pressure disturbance monitoring system was developed in 2018 and pilot-tested by the Korea Meteorological Administration (KMA). The monitoring system determines the possibility of meteotsunami occurrence based on the intensity and speed of air pressure disturbances observed from 89 meteorological stations (Kim et al., 2021). However, since there were no previous studies on temporal patterns for meteotsunami occurrence in the area, the operation of the monitoring system was limited to March–April, with reference to the timing of the strongest meteotsunami event on March 31, 2007. It is impractical for the KMA to monitor various natural hazards in real time to operate the meteotsunami monitoring system year-round, due to a limited workforce and resources. Moreover, during the test period, the real-time decision-making process was restricted to a dichotomous decision (occurrence/non-occurrence), because there was no risk-level assessment for meteotsunami occurrence. This decision-making process allows the monitoring system operator to make quick and easy decisions; however, it can cause frequent false alarms or false negatives. Therefore, specific guidelines and recommendations based on the occurrence characteristics of meteotsunami events are required for future operational efficiency and risk-level assessment. The risk-level assessment needs to be discussed to provide more accurate meteotsunami warnings for each air pressure disturbance.

The objective of this study was to quantify the occurrence frequency and characteristics of pressure-forced meteotsunami events in the eastern Yellow Sea over the past decade (2010–2019). Based on these events, the intensity, occurrence rate, and propagation of air pressure disturbances were examined and discussed for the meteotsunami warning system. In addition, local amplification in harbors was considered as an important characteristic of meteotsunamis along the eastern Yellow Sea coast.

## 2 Meteotsunami monitoring system

Meteorological station data from 2010 to 2019 were obtained from the 89 automatic weather stations (AWSs) utilized in the monitoring system (Fig. 1). We used the mean sea level pressure recorded at 1 min intervals to calculate the air pressure disturbance. Of the 89 AWSs, 17 AWSs act as beacons and are located on offshore islands along the eastern Yellow Sea. They allow for earlier observations of air pressure disturbances and, thus, for preliminary warnings. The remaining 72 AWSs, which detect the propagation (direction and speed) of air pressure disturbances, were located along the eastern coast of the Yellow Sea, including Jeju Island. Radar images covering the same area as the AWSs were used to estimate the propagation and spatial scale of the air pressure disturbances over time. In addition, 16 tide gauges were selected for use in the study, based on their data collection percentage (the percentage of successfully collected data points out of the total possible) for sea level records during the last 10 years. The data collection percentage of all the tide gauges was 72–99% and exceeded 95% at more than 10 tide gauges. The observation system in this study was divided into five latitude bands (Lat. A–E) to assess the spatial occurrence of pressure-forced meteotsunamis. To assign approximately the same number of tide gauges to each latitude band, the TaeAn (TA) and ChuJado (CJ) tide gauges were assigned to Lat. A and D, respectively. The tide gauges in Lat. A showed

lower data collection percentages relative to the other tide gauges, as 2012 was their first year of operation. The sea level data were sampled at 1 min intervals, which was equal to the sampling interval of the pressure data.

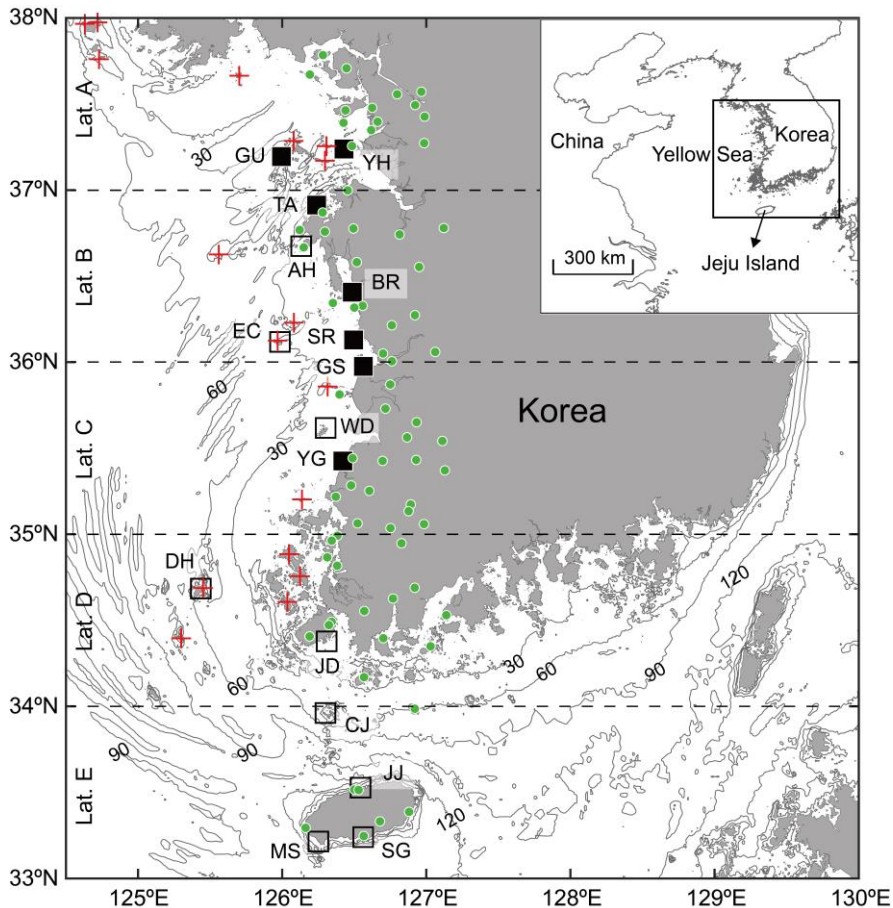

**Figure 1.** Observation system including 89 automatic weather stations (AWSs) and 16 tide gauges along the coast of the eastern Yellow Sea, with depth contours (m). Red crosses mark the 17 AWSs in the caution zone, which are outward-located
beacons of the real-time monitoring system for meteotsunami. Green circles mark the 72 AWSs in the warning zone, intended for timely warnings near the coast. Black squares indicate the 16 tide gauges: YeongHeungdo (YH), GulUpdo (GU), TaeAn (TA), AnHeung (AH), BoRyeong (BR), SeochunmaRyang (SR), EoChungdo (EC), GunSan (GS), WiDo (WD), YeongGwang (YG), DaeHeuksando (DH), JinDo (JD), ChuJado (CJ), JeJu (JJ), SeoGwipo (SG), and MoSeulpo (MS). The tide gauges were divided into five latitude bands: Lat. A (YH, GU, and TA), Lat. B (AH, BR, SR, and EC), Lat. C (GS, WD, and YG), Lat. D
(DH, JD, and CJ), and Lat. E (JJ, SG, and MS). Black empty squares represent the tide gauges located in harbors.

Meteotsunamis are initiated by traveling air pressure disturbances that change rapidly over a short period of time (Hibiya and Kajiura, 1982; Monserrat et al., 2006). Accordingly, calculating the threshold for an air pressure disturbance is a

core part of meteotsunami monitoring. This study defines an air pressure disturbance by the rate of pressure change, which is also known as the pressure tendency (Šepić et al., 2009; Šepić and Vilibić, 2011). The air pressure disturbance at every 1 min interval was calculated by the moving rate of the pressure change over 10 min, similar to the moving average method. Additionally, we tested shorter and longer time intervals for the rates. Shorter intervals are more sensitive, but from the point of view of a real-time monitoring system operation, it is necessary to consider the delay in time (approximately 10 min) for the raw pressure data observed at each AWS to be sent to the KMA. Thus, we decided on a 10 min rate. The minimum intensity of air pressure disturbances during the meteotsunami events was examined to determine which intensity can generate meteotsunamis in the eastern Yellow Sea. The referenced meteotsunami events included those that were revealed due to severe accidents and captured by the KMA real-time monitoring system in 2018 (Kim et al., 2021). Air pressure disturbances with temporal gradients greater than 0.15 hPa/min were common during the meteotsunami events (Kim et al., 2019; Bechle et al., 2015; Dusek et al., 2019). Therefore, we defined air pressure disturbances that exceeded 1.5 hPa/10 min and that could potentially generate meteotsunamis as air pressure jumps. A more detailed protocol for the monitoring system is explained in Section 3.1.

## 3 Pressure-forced meteotsunami events

### 3.1 Classification and identification of meteotsunami events

Meteotsunamis are distributed in the same frequency range as the tsunami frequency band (Monserrat et al., 2006). The sea-level oscillations observed during the meteotsunami events that resulted in accidents in the Yellow Sea (Choi et al., 2008; Choi and Lee, 2009; Eom et al., 2012; Kim et al., 2014) were also distributed in the high-frequency bands (periods of less than 2 h). To create criteria for the classification of meteotsunami events, we identified the occurrence characteristics of the destructive meteotsunami events in the eastern Yellow Sea. A meteotsunami event on April 26, 2011, was a relatively mild event compared to the more destructive meteotsunami events on 03/31/2007 and 05/04/2008, as mentioned above. However, this meteotsunami caused significant property damage to fishing boats and fish farms in the DaeHeuksando (DH) harbor (Kim et al., 2019). Figs. 2a–d shows the sea level pressure, air pressure disturbance, sea level, and high-frequency sea level during the event observed at the AWS and tide gauge located in the DH harbor (Fig. 1). In addition, the wave period was estimated based on the wavelet power spectrum when the peak-to-trough height of the high-frequency sea level was the daily maximum (Figs. 2e,f). We applied a high-pass filter with a continuous wavelet analysis based on the Morlet wavelet (Torrence and Compo, 1998). The daily maximum wave height was calculated as the largest peak-to-trough wave height in the daily data. The maximum wave height of the high-frequency sea level (Figs. 2c,d) was accompanied by a strong air pressure disturbance exceeding the intensity threshold for an air pressure jump (Figs. 2a,b). Similar to previous research findings, the meteotsunami wave heights were detected sequentially in multiple tide gauges along the propagation path of the air pressure disturbance (Kim et al., 2019; Kim et al., 2020; Kim and Woo, 2021), which indicates a resonant effect between the propagating air pressure disturbance and long ocean waves. Local amplification was observed around the range of the resonant periods of the DH harbor

(Figs. 2e,f). Thus, the following are common characteristics of the pressure-forced meteotsunami during destructive events in the eastern Yellow Sea:

- similar timing of occurrence between the air pressure jumps and high-frequency sea level,
- spread of the maximum wave heights to more than three tide gauges, and
- strong amplification by resonant periods at harbor tide gauges.

After the accident, a stronger group of pressure jumps (Fig. 2a) was detected; however, the sea-level oscillations had much smaller amplitudes (Fig. 2c). Thus, the favorable conditions for meteotsunami occurrence could include not only the intensity of the air pressure jump, but also other characteristics of the jump or wave interference conditions.

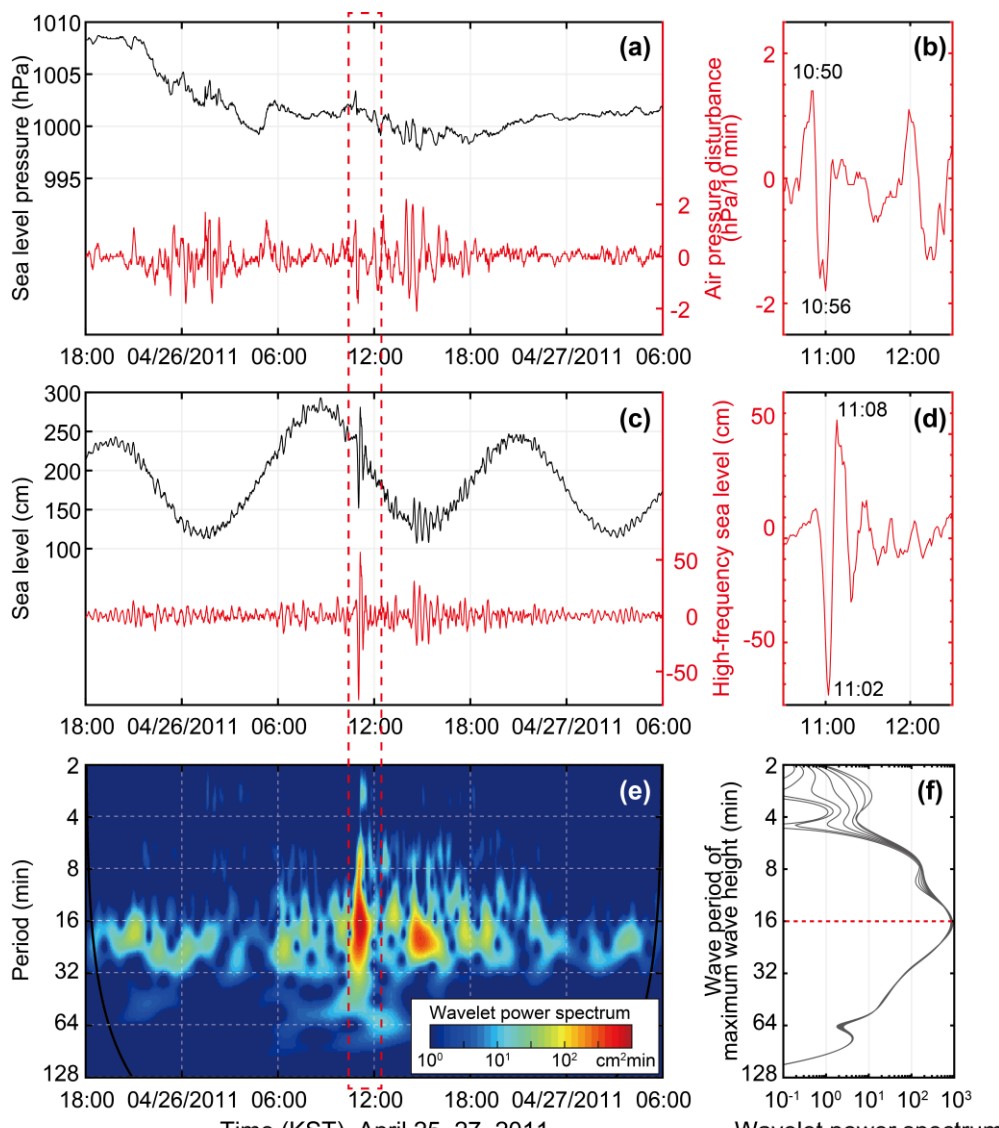

**Figure 2.** Characteristics of the pressure-forced meteotsunami from the DaeHeuksando (DH) harbor during the April 26, 2011, meteotsunami event: (a) 36 h record of sea level pressure and air pressure disturbances, (b) 2 h record of air pressure disturbances for clarity, (c) 36 h record of sea level and high-frequency sea level, (d) 2 h record of high-frequency sea level for clarity, (e) wavelet power spectrum for the high-frequency sea level, and (f) wave period estimated from the maximum wavelet power spectrum (red dashed line) during the maximum wave height. The time of the destructive meteotsunami accident is highlighted by the red dashed box.

Several attempts have been made to determine what should be considered a meteotsunami event. To date, the threshold criteria for a meteotsunami event are based on the wave amplitude, height, and energy of the high-frequency sea level, as follows:

– an absolute threshold criterion of wave height exceeding 5–100 cm in any given region (Linares et al. 2016; Pattiaratchi and Wijeratne, 2014; Pellikka et al., 2020; Rabinovich and Monserrat, 1996; Šepić et al., 2012, Williams et al., 2021);

– a relative threshold criterion of wave amplitude exceeding two or three sigma (Kim et al., 2016a, 2019);

– a combined threshold criterion of relative and absolute wave heights exceeding four sigma and a minimum absolute wave height that is specified for a given region (Monserrat et al., 2006);

– a combined threshold criterion of relative wavelet energy and absolute wave heights using a wavelet energy threshold greater than six sigma and a minimum absolute wave height greater than 20 cm (Dusek et al., 2019).

During the destructive meteotsunamis in the eastern Yellow Sea, the levels of background noise at period bands of less than 1 h differed noticeably between sites (Kim and Woo, 2021; Kim et al., 2016b). The absolute threshold criterion caused biased meteotsunami events only at particular sites with large background noises, but the relative threshold criterion classified even several minor events as the meteotsunami events. The combined threshold criterion has advantages in filtering out numerous minor events because this approach restrictively considers only potentially hazardous events as "meteotsunamis" by using both criteria (Monserrat et al., 2006). Accordingly, we used the combined threshold criterion that uses a relative wave height threshold greater than four sigma and a minimum absolute wave height greater than 20 cm as the meteotsunami intensity threshold. This intensity threshold was selected through prototyping with known meteotsunami events since 2010.

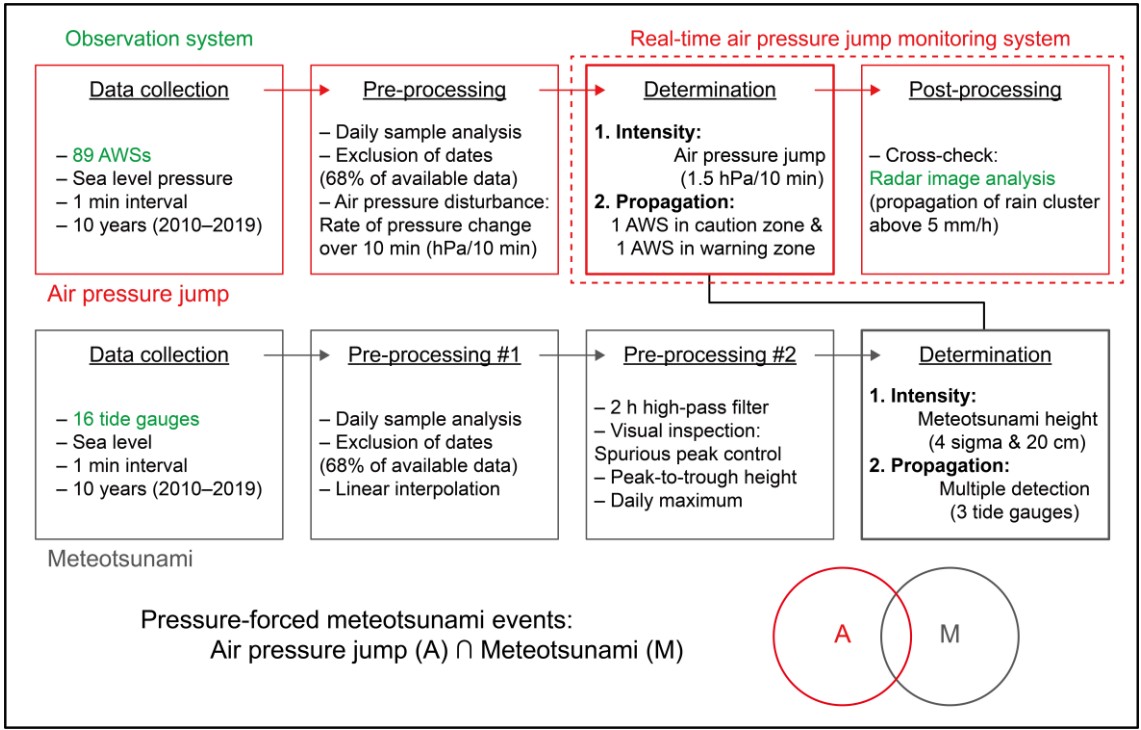

**Figure 3.** Process flow diagram showing classification of pressure-forced meteotsunami events.

The classification of air pressure jump events was performed through data collection, pre-processing, determination, and post-processing (Fig. 3). Each daily sample with a data collection percentage of less than 68% (one sigma) was excluded in the pre-processing step. When analyzing air pressure jump events in the pre-processing and determination steps, we used the following monitoring system protocols (from Kim et al., 2021):

–    the observation system utilized 89 AWSs, of which 17 AWSs were in the caution zone (red crosses in Fig. 1) and 72 AWSs were in the warning zone (green circles in Fig. 1);

–    intensity threshold of air pressure jump exceeding the rate of pressure change of 1.5 hPa/10 min; and

–    the propagation of air pressure jumps from at least one AWS in the caution zone to at least one AWS in the warning zone.

As a post-processing step, the propagation patterns of air pressure jumps, which were estimated from multiple AWSs, were cross-checked by visual inspection with the evolution of rain clusters on radar images. Only the dates when the two propagation patterns matched were classified as air pressure jump dates.

The pressure-forced meteotsunami events were classified based on two high-frequency sea-level characteristics, which were the co-occurrence with an air pressure jump and propagation of the daily maximum wave heights, as explained

above. As with the air pressure jump events, each daily record of sea level with a high rate of missing data was removed in the first pre-processing step. Linear interpolation was used to fill in missing data with short gaps. Note that a wavelet filter, which is useful for isolating localized peaks and non-periodic signals (Torrence and Compo, 1998), was used to extract the high-frequency component of sea-level oscillations of less than 2 h. Spurious peaks due to the missing data were eliminated based on visual inspection. Then, the maximum wave height of the high-frequency sea level for the quality-controlled daily sample can be obtained. If the maximum wave heights exceeded the intensity of the meteotsunami threshold at more than three tide gauges (Fig. 1) on the same date, that date was considered a meteotsunami event. Thus, we can classify the dates when an air pressure jump and a meteotsunami occurred together as pressure-forced meteotsunami events.

**Table 1.** Maximum wave height (cm) of the 42 pressure-forced meteotsunami events. The known events since 2010 are marked by a superscript. The event dates are presented in Month/Day/Year format. The intensities exceeding the meteotsunami limit are denoted by bold text, and the highest intensity for each event is denoted by underlined text. Dash marks in the table indicate a date with less than 68% of available daily data at each tide gauge.

| Date | Lat. A | | | Lat. B | | | | | Lat. C | | Lat. D | | | Lat. E | | |
|---|---|---|---|---|---|---|---|---|---|---|---|---|---|---|---|---|
| | YH | GU | TA | AH | BR | SR | EC | GS | WD | YG | DH | JD | CJ | JJ | SG | MS |
| 02/10/2010 | – | – | – | **30.6** | 17.8 | – | – | **42.2** | – | – | **43.1** | 24.6 | 33.5 | – | – | **59.8** |
| 02/11/2010 | – | – | – | **28.4** | 11.6 | – | – | 17.9 | – | – | **40.9** | 27.1 | 49.8 | – | 53.2 | **73.9** |
| 03/01/2010 | – | – | – | **28.3** | – | – | **34.9** | – | – | – | **37.7** | 24.7 | **51.5** | – | – | 39.4 |
| 03/03/2010 | – | – | – | 11.4 | – | – | 15.3 | – | – | – | 21.6 | 17.2 | **21.0** | – | **53.3** | 37.2 |
| 03/22/2010 | – | – | – | 16.4 | – | – | 13.9 | 10.1 | – | – | **31.5** | **36.0** | 31.6 | – | 19.7 | 24.1 |
| 04/21/2010 | – | – | – | **30.3** | – | – | **30.2** | **33.3** | – | – | 24.1 | 12.2 | 16.7 | – | – | 14.9 |
| 05/24/2010 | – | – | – | – | – | – | 19.9 | 10.5 | – | – | **78.4** | 28.1 | 43.1 | – | 57.5 | 38.7 |
| 04/26/2011[a] | – | – | – | 21.3 | 11.2 | – | **39.6** | 18.0 | – | – | **132.1** | – | 41.8 | – | – | **46.3** |
| 04/30/2011 | – | – | – | **36.1** | 20.5 | – | **41.3** | 25.9 | – | – | **43.1** | 16.3 | 20.2 | – | – | 38.4 |
| 05/21/2011 | – | – | – | **37.0** | – | – | **46.2** | 30.6 | – | – | 24.6 | 6.9 | 8.4 | – | – | 12.0 |
| 06/08/2011 | – | – | – | **36.5** | – | – | **48.9** | 36.8 | – | – | 35.6 | 7.7 | 11.9 | – | 27.7 | 16.4 |
| 04/03/2012 | 9.3 | 6.1 | 12.5 | 13.9 | 8.3 | 15.1 | 13.7 | 11.7 | – | – | 27.9 | 26.7 | – | – | 42.9 | **44.4** |
| 07/05/2012 | – | 10.1 | 10.0 | – | **21.8** | 29.1 | 29.7 | **31.4** | 24.3 | – | 19.4 | 8.2 | – | – | 9.4 | 17.7 |
| 07/06/2012 | – | 11.3 | 19.8 | – | 15.7 | 14.3 | **25.7** | 20.5 | 20.3 | – | 17.4 | 10.7 | 10.7 | – | 10.5 | 19.3 |
| 01/20/2013 | **20.8** | 14.9 | **26.3** | 23.6 | – | 12.7 | 18.2 | – | 19.4 | – | 21.7 | 12.0 | 13.1 | – | 15.9 | 19.2 |
| 02/03/2013 | 6.8 | 7.8 | 6.0 | 15.6 | – | 14.4 | **21.2** | – | 29.4 | – | 36.0 | 27.7 | 23.6 | – | 22.3 | **61.0** |
| 03/10/2013 | 16.3 | – | 9.0 | – | 5.5 | 13.2 | 17.5 | – | 23.7 | – | **31.3** | 21.6 | 18.2 | – | 18.1 | 29.5 |
| 04/14/2013 | 10.3 | 15.7 | 21.5 | – | 12.1 | **60.0** | **60.7** | 19.1 | 49.6 | – | 34.2 | 23.1 | **21.0** | – | – | 26.0 |
| 04/29/2013 | 13.3 | 14.0 | 15.9 | **22.3** | 8.5 | **25.7** | **39.7** | 14.8 | **33.1** | – | 21.9 | 8.9 | 8.9 | – | – | 11.6 |
| 07/03/2013 | 8.7 | 6.5 | 7.5 | **29.5** | – | 21.7 | 17.4 | 15.7 | **42.5** | – | 34.6 | 10.1 | 15.8 | – | 10.8 | 17.1 |
| 08/10/2013 | **25.8** | – | 19.5 | – | – | 17.0 | **23.0** | 20.4 | 25.1 | – | – | 7.2 | – | – | 7.1 | 5.5 |
| 04/04/2015[b] | 10.2 | 16.1 | 17.7 | **48.5** | – | 29.5 | – | – | 20.5 | 35.3 | 35.8 | 20.1 | 21.7 | 17.7 | 29.0 | **40.1** |
| 05/12/2015 | – | **33.5** | 13.7 | **31.4** | – | 29.0 | – | – | 32.9 | 31.6 | **34.5** | 18.6 | 23.6 | 20.7 | 39.2 | 19.6 |
| 06/13/2015 | **21.0** | 18.8 | **24.1** | **38.4** | – | 9.8 | 12.2 | – | 15.1 | 22.3 | 15.2 | 13.5 | 9.8 | 9.3 | – | 20.9 |
| 08/11/2015 | 5.2 | – | 4.0 | 11.2 | – | 13.6 | 11.6 | 4.3 | 18.2 | 32.0 | 17.5 | 12.8 | 10.1 | 31.8 | 12.1 | **33.2** |
| 04/16/2016[b] | 5.0 | – | 6.5 | – | 5.2 | – | 11.5 | – | 11.9 | 21.1 | 20.2 | 25.4 | 27.6 | – | **52.8** | 25.8 |
| 06/15/2016 | 12.9 | 20.5 | 13.9 | **34.4** | 11.1 | 16.3 | **22.7** | – | 28.0 | 30.5 | 32.6 | – | 10.9 | – | 11.8 | – |
| 06/24/2016 | 11.9 | 11.3 | 14.7 | **36.7** | – | 18.6 | **26.3** | 12.8 | 29.8 | 44.3 | **45.5** | 16.5 | – | 12.4 | 11.7 | 22.2 |
| 04/18/2017 | 9.2 | 15.1 | – | – | 5.0 | 15.2 | 20.7 | – | 21.8 | 36.9 | – | 6.1 | – | **41.3** | 18.1 | 11.1 |
| 03/04/2018[c] | 13.4 | – | 13.2 | **33.0** | – | – | 34.3 | 45.0 | 49.7 | **67.3** | 48.4 | 25.0 | – | – | 17.7 | **34.4** |
| 04/10/2018[c] | 15.6 | – | 10.6 | **38.2** | – | – | 29.6 | – | 22.2 | – | – | 8.0 | 8.6 | – | 10.5 | – |
| 05/16/2018 | 11.1 | 13.2 | 11.2 | **32.0** | – | 21.2 | **22.4** | – | 18.8 | – | 16.7 | 7.2 | – | – | 7.1 | 9.3 |

| | | | | | | | | | | | | | | | | |
|---|---|---|---|---|---|---|---|---|---|---|---|---|---|---|---|---|
| 05/17/2018[b] | 13.5 | **24.0** | **21.2** | **35.9** | – | 15.5 | 17.0 | 15.4 | **25.6** | – | **31.7** | 8.6 | 15.1 | – | 10.7 | – |
| 06/09/2018 | – | – | – | **22.2** | – | **24.6** | **28.4** | – | **32.9** | – | – | 11.7 | 15.0 | – | 11.5 | – |
| 10/06/2018 | 9.8 | – | 5.6 | 17.4 | 5.2 | 8.1 | 10.8 | 9.7 | 11.8 | 13.6 | 10.4 | **21.9** | **25.9** | **44.5** | 40.0 | 31.3 |
| 03/20/2019[b] | 7.9 | 13.2 | 14.8 | **29.1** | – | **25.5** | – | – | – | **54.2** | **66.4** | 28.7 | 29.8 | 22.2 | 25.7 | – |
| 03/30/2019 | 7.6 | **35.1** | 12.3 | 21.0 | – | 11.5 | 12.0 | – | **20.8** | 26.5 | **29.1** | 10.3 | 15.2 | 11.9 | 23.6 | 18.2 |
| 04/07/2019 | 12.8 | 6.5 | 5.7 | – | – | 16.3 | 18.1 | – | 14.8 | 20.2 | 24.0 | **22.1** | **30.3** | **22.4** | 30.1 | **77.7** |
| 04/09/2019[b] | 11.0 | 19.4 | 14.7 | **31.1** | – | 19.5 | **28.3** | – | – | 26.4 | 16.2 | 15.8 | **21.6** | 16.2 | 23.2 | – |
| 06/06/2019 | 10.7 | 8.4 | 8.9 | – | – | 16.8 | **24.1** | **24.8** | – | 25.8 | 23.3 | – | **21.2** | – | 11.8 | 13.8 |
| 09/07/2019 | 11.2 | – | – | **45.6** | – | **20.5** | **31.4** | 12.8 | – | 23.4 | 13.7 | **34.5** | **34.3** | – | 38.8 | – |
| 11/10/2019 | 16.0 | **29.8** | **24.3** | **38.7** | – | **23.9** | **29.0** | – | **30.3** | **39.8** | 24.1 | 14.2 | – | 11.0 | 11.8 | 16.0 |

**a: destructive event, b: event revealed by KMA internal reports, c: event captured by KMA real–time monitoring system.**

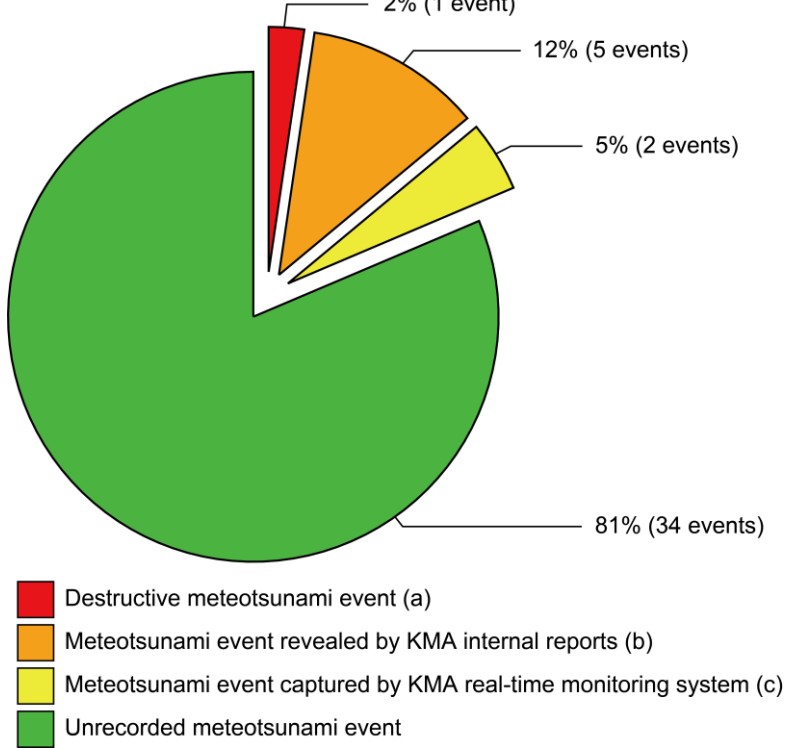

**Figure 4.** Percentage of meteotsunami event types.

Table 1 summarizes the maximum wave height at each latitude band and tide gauge during the pressure-forced meteotsunami events. To examine the validity of the classified 42 pressure-forced meteotsunami events (Table 1 and Fig. 4), we checked for the detection of the following known meteotsunami events since 2010 (Month/Day/Year):

- a destructive event on 04/26/2011;
- five events revealed by the KMA internal reports, which occurred on 04/04/2015, 04/16/2016, 05/17/2018, 03/20/2019, and 04/09/2019; and
- two events that were captured by the real-time monitoring system on 03/04/2018 and 04/10/2018.

All the known events were detected (Table 1) according to intensity and propagation thresholds (Fig. 3) that were based on the common characteristics of destructive meteotsunami events (03/31/2007, 05/04/2008, and 04/26/2011); thus, they were confirmed as reasonable. However, the percentage of unrecorded meteotsunami events between 2010 and 2019 was 81% (Fig. 4). In fact, meteotsunamis in the eastern Yellow Sea occurred more frequently than expected, thus presenting an overlooked and underrated hazard (Pattiaratchi and Wijeratne, 2015).

### 3.2 Temporal and spatial pattern of meteotsunami occurrences

The monthly distribution of the events was quantified to examine the temporal pattern of pressure-forced meteotsunami occurrences in the eastern Yellow Sea (Fig. 5a) and showed a strong seasonal trend. Seventy-one percent of meteotsunamis (30/42) occurred in March–June, which is more frequent than the average of 3.4 events per month. This four-month period also included the months of the most destructive meteotsunami events (03/31/2007, 05/04/2008, and 04/26/2011). Meteotsunami occurrences peaked in April, with nearly 29% of the total events (12/42). Meteotsunamis occurred less than once a month from September to January during the past decade. The meteotsunami strength was estimated using box and whisker plots of the statistical meteotsunami wave height per month (Fig. 5b). The wave height statistics in February revealed the largest range (whisker), interquartile range, and median. Moreover, not only did meteotsunamis occur frequently from March to June, but also destructive meteotsunami heights (circled outliers) due to strong amplification were significantly higher than in other months. The overestimated wave height statistics from September to November may be biased because of the small number of events per month. Thus, a real-time monitoring system in the eastern Yellow Sea (Kim et al., 2021) should be operated intensively in the spring season based on peak meteotsunami seasonality and strength.

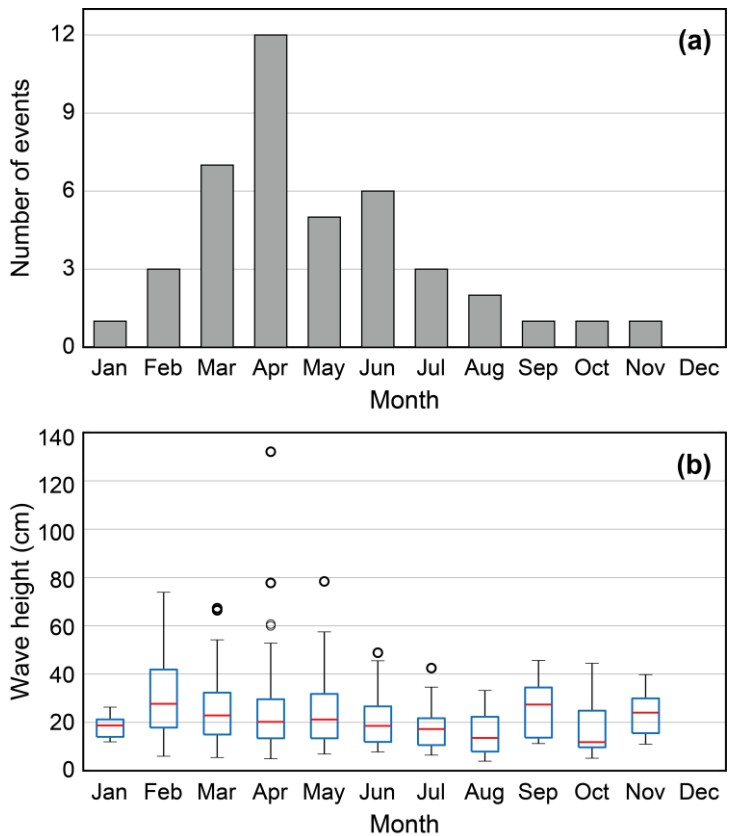

**Figure 5.** Temporal pattern of meteotsunami occurrences: (a) number of events and (b) distribution of wave height per month.

The spatial pattern of the events per year was examined to find potential hot spots where meteotsunamis were frequent within each latitude band and at each tide gauge (Fig. 6). Meteotsunamis occurred at different frequencies in each latitude band and at each tide gauge as shown in Fig. 6a. Except for 2010, most of the meteotsunami events each year occurred in Lat. B–D. Interestingly, the deviation in the number of events for each tide gauge was distinct, even within the same latitude band. The total number of events per tide gauge is shown in Fig. 6b; the geometric features of the basins exceeding the average number of occurrences among the 16 tide gauges (11.8, 188/16) are labeled. Meteotsunamis most frequently occurred above the average per latitude band at the AH, WD, DH, and MS harbor tide gauges. Frequent meteotsunamis also occurred at the EC harbor tide gauge; the EC harbor had a large quality factor (Q-factor), which is an aspect ratio between the length and width of the harbor (Rabinovich, 2009). The Q-factor determines the amplification of wave height when pressure-forced meteotsunamis propagate toward harbors (Monserrat et al., 2006). Overall, the spatial pattern of meteotsunami occurrences along the eastern Yellow Sea coast has been characterized as "harbor meteotsunamis" (Rabinovich, 2020) which result in destructive harbor oscillations throughout multiple harbors.

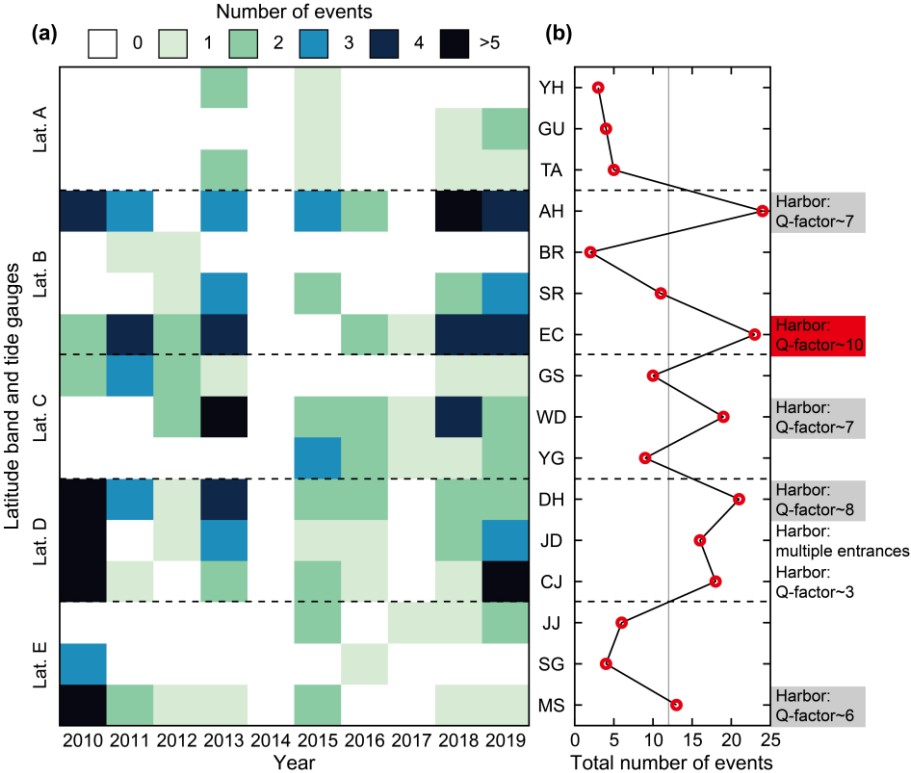

**Figure 6.** Spatial pattern of meteotsunami occurrences: (a) number of events per year and (b) total number of events. Gray lines indicate average occurrences at the tide gauges (11.8). Geometric features are presented for the basins that have greater than average occurrences at their tide gauges. The tide gauges where the most events occurred at each latitude band and located in the harbor with the largest quality factor (Q-factor) are highlighted in gray and red, respectively.

## 4 Occurrence characteristics of meteotsunamis

### 4.1 Extreme (widespread) meteotsunami events

Out of the 42 classified meteotsunami events (Table 1), extreme meteotsunami events were classified based on more hazardous conditions. The characteristics favorable for extreme meteotsunami generation are vital to consider when developing a meteotsunami warning system. Meteotsunamis that spread over a large area can be dangerous because the eastern Yellow Sea coast is in a harbor meteotsunami-dominated environment characterized by many harbors along the long and complicated coastline. The long ocean waves forced by the propagating pressure jump line can generate widespread and destructive harbor meteotsunamis, caused by local amplification in multiple harbors (Kim and Woo, 2021). During the monitoring system pilot operation, the meteotsunamis that were amplified by the Proudman resonance and propagated on a wider spatial scale were more hazardous than the meteotsunamis with a local scale (Kim et al., 2021). Therefore, the spatial

scale can be considered as a parameter for meteotsunami severity from the perspective of monitoring system operation on the
eastern Yellow Sea coast. In this study, we classified 11 extreme (widespread) events (Table 2), from among the 42 pressure-forced meteotsunami events, based on the following threshold criteria:

–    at least six tide gauges where the meteotsunami occurred, which is twice the propagation threshold for meteotsunami (Fig. 3), and

–    greater than 50% occurrence rate, which is the ratio between the number of tide gauges where the meteotsunami
occurred to the total number of tide gauges available during the event.

The average intensity was calculated by averaging the air pressure jump and meteotsunami intensity at the AWSs or tide gauges where pressure-forced meteotsunami occurred. The occurrence rate was the percentage of observation points exceeding the intensity threshold out of the total observation points satisfying the percentage of daily sample collection (Fig. 3) at each event date.

**Table 2.** Average intensity and occurrence rates for air pressure jumps and meteotsunamis during 11 extreme (widespread) meteotsunami events. The event dates are presented in Month/Day/Year format.

| Date | Air pressure jump | | | Meteotsunami | | |
|---|---|---|---|---|---|---|
| | Average intensity (hPa/10 min) | Detected/Total AWSs | Occurrence rate (%) | Average wave height (cm) | Detected/Total tide gauges | Occurrence rate (%) |
| 02/10/2010 | 1.8 | 28/87 | 32 | 36.0 | 6/7 | 86 |
| 02/11/2010 | 2.1 | 28/87 | 32 | 37.9 | 6/8 | 75 |
| 03/01/2010 | 1.7 | 46/86 | 53 | 36.1 | 6/6 | 100 |
| 04/30/2011 | 2.6 | 40/86 | 47 | 30.2 | 6/8 | 75 |
| 02/03/2013 | 2.5 | 29/88 | 33 | 22.7 | 6/12 | 50 |
| 04/14/2013 | 1.7 | 27/88 | 31 | 29.4 | 7/12 | 58 |
| 04/04/2015 | 2.7 | 49/88 | 56 | 26.3 | 8/13 | 62 |
| 05/12/2015 | 1.7 | 12/89 | 13 | 27.4 | 8/12 | 67 |
| 03/04/2018 | 2.6 | 32/89 | 36 | 34.7 | 8/11 | 73 |
| 03/20/2019 | 2.5 | 47/88 | 53 | 28.9 | 7/11 | 64 |
| 11/10/2019 | 2.1 | 34/87 | 39 | 23.8 | 7/13 | 54 |

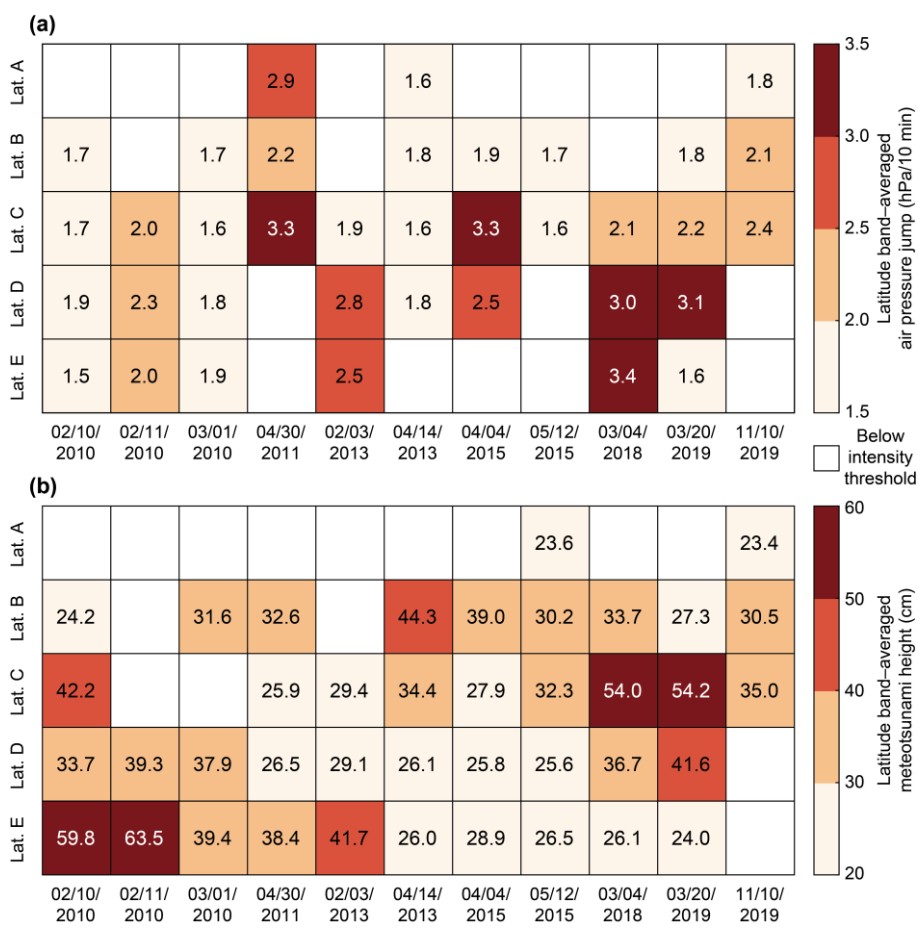

**Figure 7.** Latitude band-averaged intensity of extreme meteotsunami events: (a) air pressure jump and (b) meteotsunami.

The latitude band-averaged intensity heatmap was examined to compare the spatial relationships of the air pressure jump to the meteotsunami height for the extreme meteotsunami events, as shown in Fig. 7. The latitude bands where the air pressure jump and meteotsunami were below the intensity threshold are shown as blanks in the heatmap. The intensity ratio of meteotsunamis to air pressure jumps ranged from 7.7–39.9, showing different intensity ratios according to event and latitude band. Interestingly, the latitude bands with the maximum intensity air pressure jumps did not match those with the maximum meteotsunami heights. In addition, latitude band-averaged wave heights exceeding the meteotsunami intensity threshold were detected even in latitude bands below the air pressure jump intensity threshold. These discrepancies suggest that the intensity of the air pressure jump alone is insufficient to explain the favorable conditions for meteotsunami occurrence (Šepić and Rabinovich, 2014).

## 4.2 Propagation of the air pressure jump

When a strong air pressure jump with large spatial coverage propagates to multiple stations over several hours, widespread and significantly amplified meteotsunamis can be generated (Hibiya and Kajiura, 1982; Rabinovich et al., 2021; Šepić et al., 2012). To examine how the intensity and propagation characteristics of the air pressure jumps affect meteotsunami occurrence in the Yellow Sea, the meteotsunami event on April 4, 2015, was selected from the 11 extreme events as an example to compare the temporal and spatial occurrences of air pressure jump-meteotsunami. In this extreme event, the most hazardous air pressure jump propagated such that the average intensity of multiple AWSs was almost twice the intensity threshold, and the occurrence rate of the air pressure jump exceeded 50% (Table 2). The resultant meteotsunamis had an average intensity of 26.3 cm (Table 2) and were detected at the AH, SR, WD, YG, DH, JD, CJ, and MS tide gauges, which were located in Lat. B-E (Table 1).

The analysis of the air pressure jump characteristics that generated the meteotsunamis was based on the propagation pattern, using the arrival times and maximum intensity at each AWS (Fig. 8a). The temporal and spatial occurrence of the air pressure jump was cross-checked by utilizing the radar image corresponding to the arrival time (Figs. 8b–d). Radar images can be used to track the temporal and spatial distribution of air pressure jumps because there is a high correlation between the intensity of the air pressure disturbance and the reflectivity of the radar (Linares et al., 2016; Pellikka et al., 2020; Wertman et al., 2014). Based on the records from the tide gauges, AWSs, and radar images, we found the following characteristics of the air pressure jump during this extreme meteotsunami event:

– similar arrival timing and spatial pattern between the intensity of the rain rate above 5 mm/h and the air pressure jump above 1.5 hPa/10 min,
– propagation of the air pressure jump toward the latitude band where the meteotsunamis were detected,
– spatial scale of meteotsunamis that was similar to or slightly larger than that of the air pressure jump, and
– a discrepancy between the latitude band where the air pressure jumps were greatest and the meteotsunamis were most intense (Fig. 7), which cannot be explained by the intensity and propagation of the air pressure jump (Fig. 8).

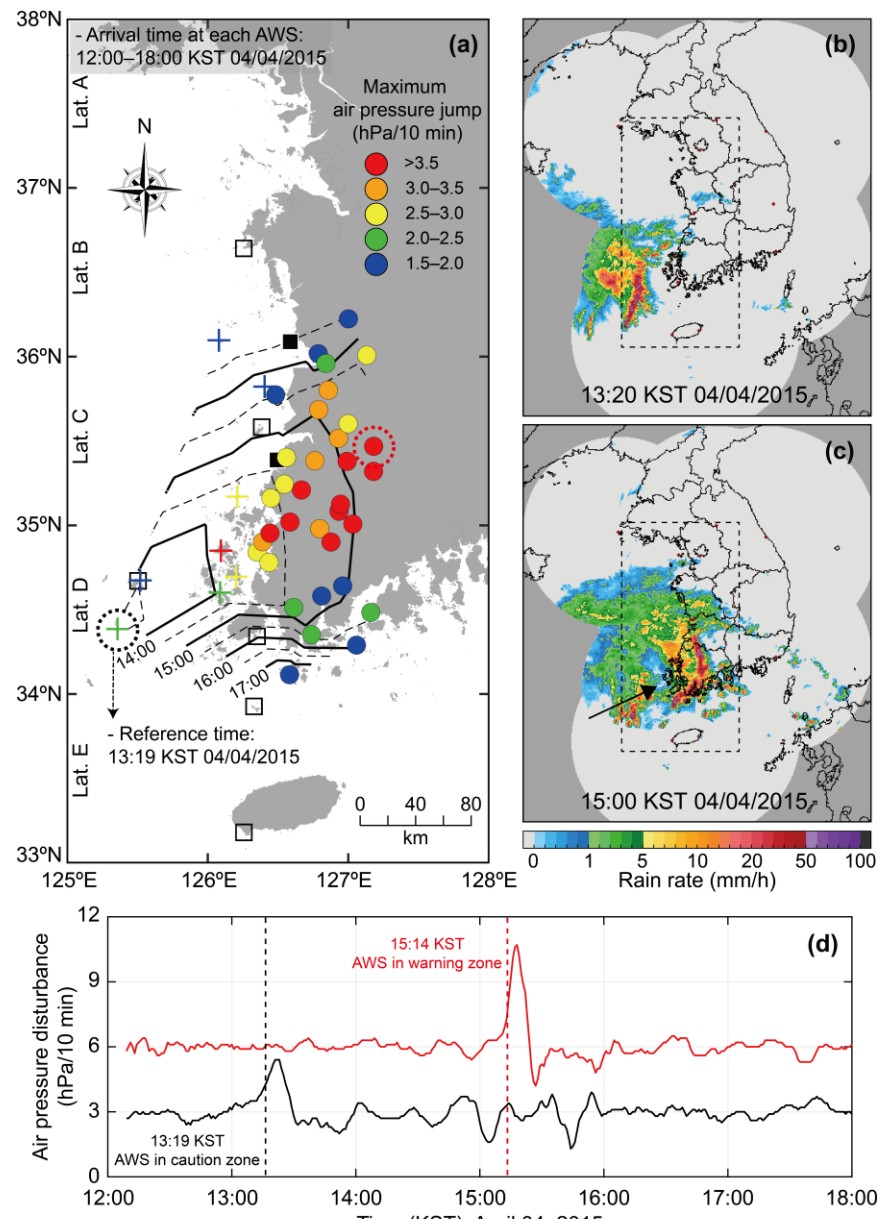

**Figure 8.** Propagation of the air pressure jump on April 4, 2015. (a) Isochrone map. Circles mark multiple AWSs where the air pressure jump arrived from 12:00 to 18:00 KST. Colors of the circles denote the maximum intensity. The reference time is the first arrival time. Thick black lines and black dashed lines indicate isochrones (1 h and 30 min) from the reference time. Black squares mark the tide gauges where the meteotsunamis arrived (see Table 1). Black empty squares are the tide gauges located in harbors. (b) Radar images at 13:20 and (c) 15:00 KST on April 4, 2015, respectively. Dashed squares indicate same area as the isochrone map. (d) The time series of the air pressure disturbances observed at two AWSs located on propagation path of the air pressure jumps (see black and red dashed circles in the isochrone map).

This study determined that the intensity of the air pressure jump was the key factor for meteotsunami generation, and favorable conditions were assessed based on the propagation characteristics (speed, direction, and occurrence rate) of the air pressure jump. The speed and direction of the air pressure jump can be favorable for the amplification of sea-level oscillations in the open sea due to the Proudman resonance (Belušić et al., 2007; Chen and Niu, 2018; Denamiel et al., 2020; Proudman, 1929; Vilibić et al., 2004; Vilibić, 2008; Šepić and Vilibić, 2011). The propagation characteristics of the air pressure jump

during the 42 meteotsunami events was estimated from an isochrone map of air pressure jump arrival at the AWSs in the same way as the analysis of the extreme event on April 4, 2015 (Fig. 8). However, it was difficult to determine the propagation due to ambiguous cases resulting from an unorganized cluster with a low occurrence rate and multiple propagation patterns. Accordingly, we selected those main directions of air pressure jumps in the isochrone map that were consistent with the propagation pattern of the rain rate in the radar images. The intensity and movement of rain rates exceeding 5 mm/h were

confirmed by visual inspection (Kim et al., 2021). The arrival time list and isochrone map of the air pressure jumps were estimated in the area where this rain rate propagated (Fig. 8). Then, the direction and speed were assessed using three data points from AWSs located in the main direction, based on the explicit formula suggested by Šepić et al. (2009). The direction $\theta$ and speed $U$ of the air pressure jumps were estimated using a triangle of AWSs with coordinates $(x_1, y_1)$, $(x_2, y_2)$, and $(x_3, y_3)$. The traveling air pressure jumps can be tracked based on the assumption that an air pressure jump does not change and

maintains a constant direction and speed during travel. The propagation pattern is expressed as follows:

$$tan\theta = a = \frac{\Delta t_{12}\Delta y_{13} - \Delta t_{13}\Delta y_{12}}{\Delta t_{13}\Delta x_{12} - \Delta t_{12}\Delta x_{13}}, \tag{1}$$

$$U = \frac{1}{\Delta t_{12}}\frac{\Delta y_{12} - a\Delta x_{12}}{\sqrt{1 + a^2}} = \frac{1}{\Delta t_{13}}\frac{\Delta y_{13} - a\Delta x_{13}}{\sqrt{1 + a^2}}, \tag{2}$$

where $\Delta t_{12}$ and $\Delta t_{13}$ are the time lags between each AWS; $\Delta x_{12}$, $\Delta x_{13}$, $\Delta y_{12}$, and $\Delta y_{13}$ are the distances between each AWS in the east-west and north-south directions, respectively.

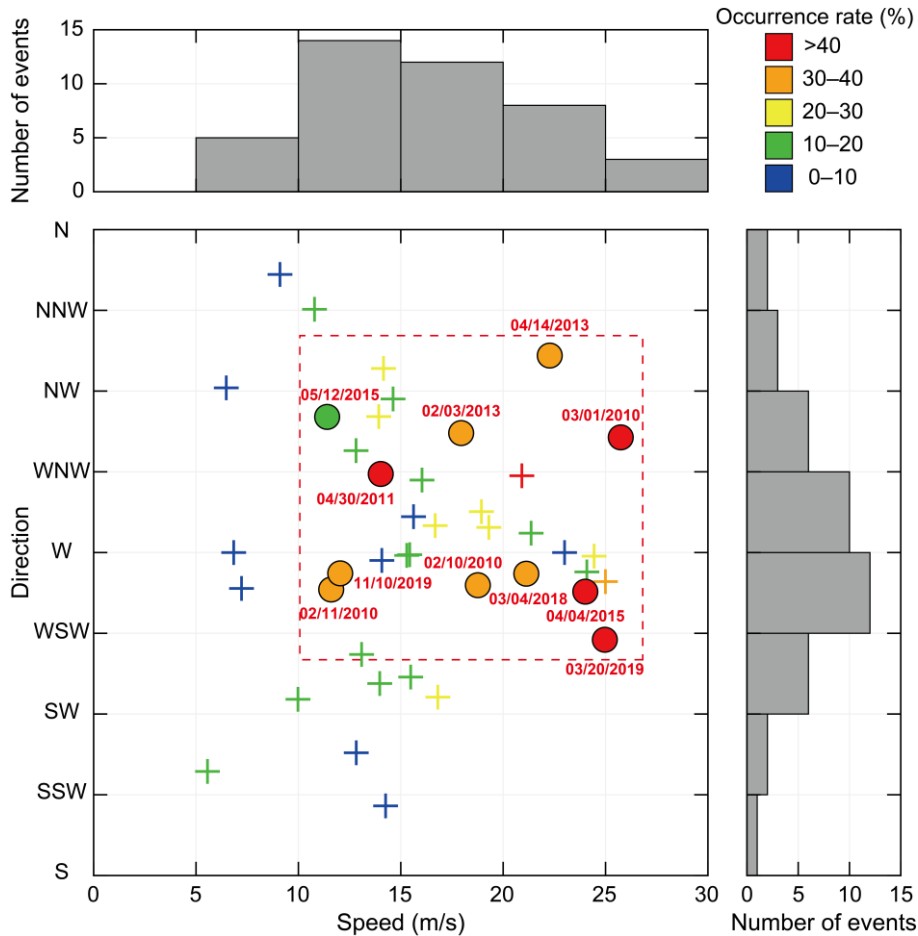


**Figure 9.** Propagation pattern of air pressure jumps during the classified meteotsunami events (speed, direction, and occurrence rate). Colors of circles and crosses indicate occurrence rate of the air pressure jump during the 11 extreme events and the remaining 31 meteotsunami events, respectively. The red dashed square in the scatterplot encloses range of speed and direction for the air pressure jumps during the extreme (widespread) meteotsunami events. The binned distributions of direction and

speed are shown.

The occurrence characteristics of the air pressure jumps during the 42 meteotsunami events are shown in Fig. 9. The scatter points show the results of the speed, direction, and occurrence rate analysis. The binned distributions indicate the dominant speed and direction, respectively. Extreme (widespread) meteotsunami events are highlighted with a circle marker and red text. The propagation patterns of the air pressure jumps were distributed in the N–S direction at speeds of 5–30 m/s.

More than 50% of the events occurred in the WNW–WSW directions with speeds of 10–20 m/s. In particular, the speed and direction of the air pressure jumps during the extreme events were locally distributed in the NNW–SW direction at speeds of 11–26 m/s (red dashed box in Fig. 9), corresponding to the Proudman resonant speed range in the eastern Yellow Sea at a

shallow water depth of 15–90 m (Fig. 1). As mentioned above, extreme meteotsunami events were defined as widespread meteotsunami events, without considering the occurrence rate of the air pressure jump. In most extreme events, air pressure

jumps at large spatial scales were recorded at more than 27 out of 89 AWSs (30%) (Table 2 and Fig. 9). Therefore, there were spatial connections between meteotsunamis and air pressure jumps. We identified specific occurrence characteristics of air pressure jumps during the extreme events in the eastern Yellow Sea, although there was the limitation that the period and start location of the air pressure jumps were not considered (Denamiel et al., 2020).

## 4.3 Local amplification in harbors

The intensity and propagation characteristics of the air pressure jumps were not sufficient to explain the intensity of the meteotsunamis on the pressure-forced meteotsunami dates. Local factors can be decisive in forecasting the severity of meteotsunamis in the eastern Yellow Sea, because the coastline is long, complicated, and has many islands with harbors. We identified local amplification in harbors as a possible reason for this discrepancy in intensity (Fig. 7), based on the harbor meteotsunamis-dominated environment (Fig. 6b). Local amplification inside a harbor occurred when the period bands of

incoming pressure-forced long waves from the open sea were similar to the resonant periods of the harbor (Rabinovich, 2009). Thus, local amplification at all tide gauges was assessed by estimating the dominant periods of the maximum wave heights during the classified events. First, the wavelet transform of the high-frequency sea level at each tide gauge was performed for each meteotsunami event (Fig. 2e). Then, the dominant wave period with the maximum power spectrum was estimated in the wavelet domain when the meteotsunami wave height was the daily maximum in the time domain (Fig. 2f).

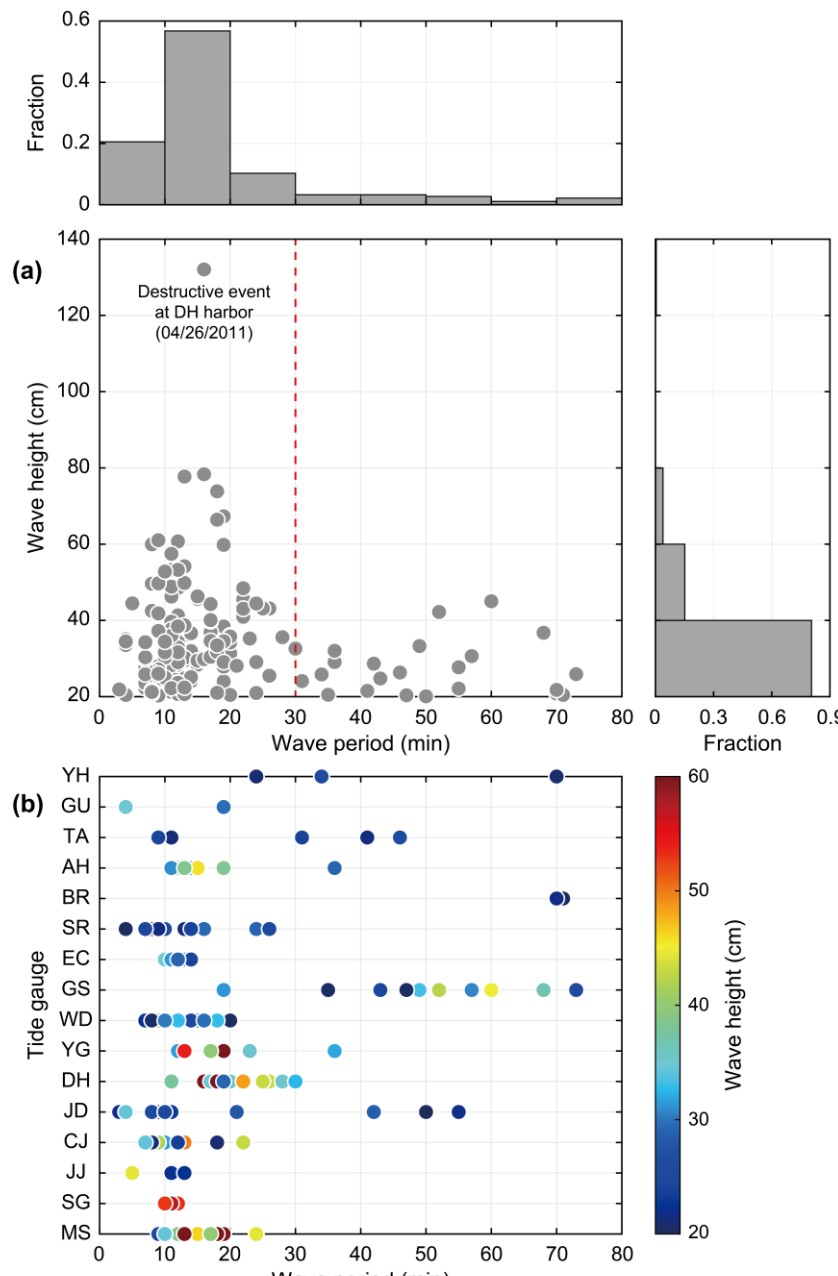

**Figure 10.** (a) Scatterplot of wave period and wave height of the meteotsunami events. The binned distributions of wave height and period are shown. The distribution of the wave period was divided into two groups by the red dashed line (30 min). (b) The dominant wave period of meteotsunami wave heights at each tide gauge. Colors of circles denote wave height.

Fig. 10 shows the distributions of wave heights and period bands for meteotsunamis along the Yellow Sea coast. Most meteotsunami heights (81%) were between 20 and 40 cm. The maximum wave height for the destructive meteotsunami events


was recorded at the DH harbor. Additionally, 88% of the meteotsunamis had dominant period bands of less than 30 min; specifically, 57% of the wave periods were between 10 and 20 min. The dominant period bands of the top 19% of the meteotsunami wave heights (>40 cm) were also less than 30 min. To identify the tide gauges at which wave period bands of less than 30 min primarily occurred, we analyzed the dominant period bands of the meteotsunami wave heights at each tide gauge (Fig. 10b). As with the meteotsunamis frequently occurring at harbor tide gauges (Fig. 6), strong meteotsunami wave heights (10–20 min periods) were also mainly observed at the harbor tide gauges. In addition, the wave heights for the events were significantly amplified at the harbor tide gauges compared those recorded at other tide gauges (Table 1). Conversely, relatively weak wave heights, with longer periods of 30–80 min and without any dominant period bands, were commonly recorded at the YH, GU, TA, BR, and GS tide gauges, which were not located in a harbor. Local amplification due to harbor resonance appears to be responsible for this discrepancy in intensity (Fig. 7). Even before 2010, meteotsunami-induced harbor seiches (i.e., harbor meteotsunamis) at multiple tide gauges were reported in the eastern Yellow Sea, generating strong amplification with resonant periods of less than 30 min (Kim and Woo, 2021; Kim et al., 2016b).

## 5 Discussion and conclusions

We classified the 42 pressure-forced meteotsunami events that occurred on air pressure jump-meteotsunami dates by using the long-term pressure and sea level data between 2010 and 2019. A distinct distribution of meteotsunami occurrences by year was not found in this study. However, seasonal factors (Fig. 6a) were related to local climatology (Vilibić et al., 2018; Williams et al., 2021). Of the classified meteotsunami events, 71% (30/42) occurred between March and June, during the spring to early summer in the Northern Hemisphere (Fig. 5a). Interestingly, these seasonal patterns were similar to those in Lake Michigan, which occur in the late spring and early summer (Bechle et al., 2015). Additionally, the peak meteotsunami seasonality in the Mediterranean Sea was found to be between June and August (Rabinovich and Monserrat, 1996; Šepić et al., 2012). These seasonal patterns of event distribution indicate that the observed events are related to atmospheric conditions and processes. The meteotsunami seasonality in the eastern Yellow Sea may be due to three synoptic weather types, which are caused mainly by the following characteristics of low-pressure systems (Kim et al., 2016a): (i) the frequent passage of low-pressure systems over the Yellow Sea, increasing potential atmospheric instability; (ii) the stagnation of low-pressure systems by a blocking high in the North Pacific; and (iii) the movement by the Westerlies of low-pressure systems that developed in the highlands (Mongolian and Tibetan plateaus) to the lowlands. The synoptic weather types in the spring season were characterized by a low-pressure system accompanying a strong cold front passing through the Yellow Sea. In other regions (especially in Europe), meteotsunami seasonality is associated with specific synoptic conditions, such as a low-pressure system at the surface, a horizontal temperature front at 850 hPa, and advection by jet stream winds at 500 hPa (Ozsoy et al., 2016; Šepić et al., 2012; Šepić et al., 2016; Vilibić et al., 2018).

There were exceptional cases other than the pressure-forced meteotsunami events classified in this study. One such case was a wind-dominated event on September 7, 2019 (Table 1), which was characterized by sudden and large changes in

wind gust speed (5–10 m/s), such as those with typhoon-induced meteotsunamis (Anarde et al., 2021; Shi et al., 2020). The cases in which the air pressure jump threshold was not satisfied but the meteotsunami still occurred may also correspond to

wind-dominated meteotsunami events. The contribution of air pressure and wind to meteotsunami generation can differ by event and region (Linares et al., 2016); however, air pressure disturbances typically play a much larger role than wind forcing (Vilibić et al., 2005). Conversely, the cases in which strong air pressure jumps were detected at multiple AWSs, but meteotsunamis were below the intensity and propagation thresholds, may be due to interferences from interactions between meteotsunamis and tides (Choi et al., 2014) or waves near the coast (Linares et al., 2019). Thus, false alarms can be reduced

by filtering out these exceptional cases as much as possible in the monitoring system. Therefore, it is vital to study the contributions of tide-, wind-, and wave-generated meteotsunami interferences.

Kim et al. (2021) developed and pilot-operated the meteotsunami monitoring/warning system, by considering the characteristics of air pressure jumps that are favorable to meteotsunami generation. In this work, the mechanisms of meteotsunami generation were partially explained by the occurrence characteristics (intensity, propagation, and occurrence

rate) of the air pressure jumps (Fig. 9). However, we found a discrepancy in intensity (Fig. 7) between the air pressure jumps and meteotsunamis within each latitude band during extreme meteotsunami events (Table 2); therefore, other mechanisms were considered. The intensity discrepancy was primarily due to local amplification at multiple harbors after the coupled-mode propagation of the air pressure jump and long ocean waves (Monserrat et al., 2006; Rabinovich, 2009). Over the past decade, the most frequent (Fig. 6) and locally amplified harbor meteotsunamis, corresponding to the top 10% of meteotsunami wave

heights, were dominant in period bands of less than 30 min (Fig. 10). These period bands correspond to the resonant periods of the harbors in the eastern Yellow Sea (Kim and Woo, 2021). This suggests that local amplification due to internal resonance occurred in multiple harbors. Therefore, it is essential to consider harbor resonance in the current monitoring/warning system, which uses only the occurrence characteristics of the air pressure jumps.

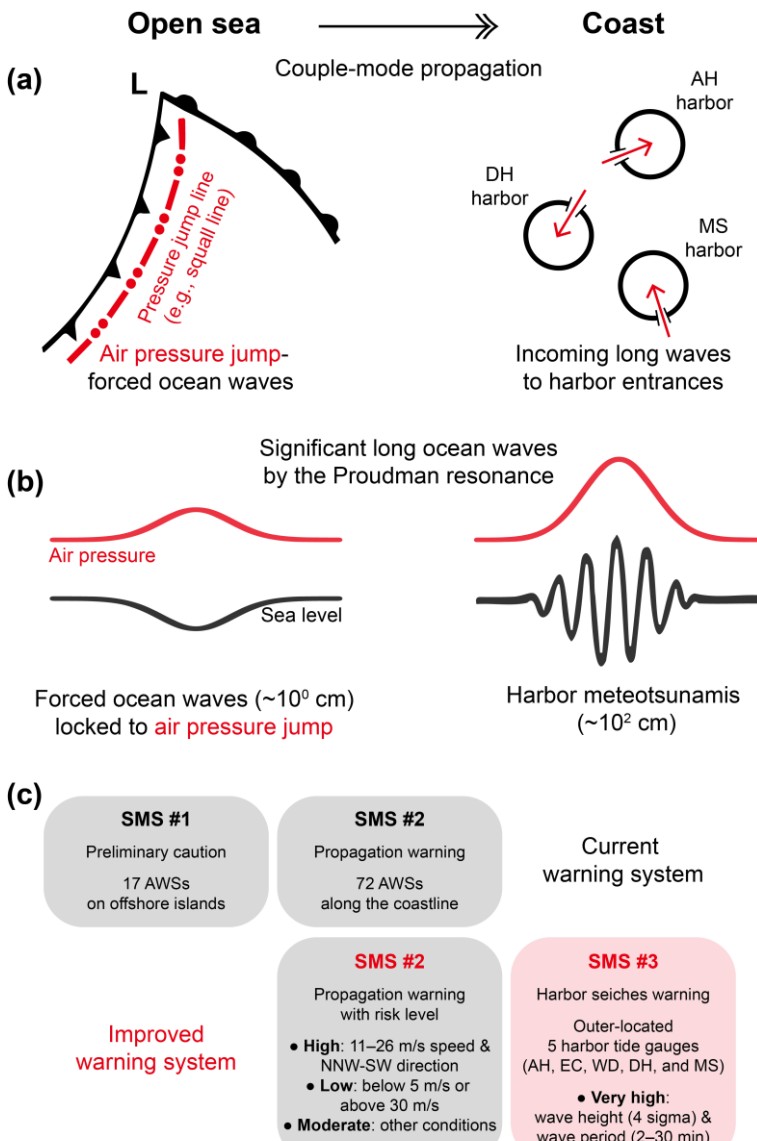

**Figure 11.** (a) Schematic diagram of harbor meteotsunamis, (b) the amplification process of ocean waves locked to air pressure jumps in the time domain, and (c) the meteotsunami warning system (as-is and to-be) in the eastern Yellow Sea.

Figs. 11a,b shows a schematic diagram illustrating harbor meteotsunamis at multiple harbors with different geometric features (e.g., entrance direction and width). A pressure jump line (e.g., squall line) traveling in the open sea can produce forced ocean waves (~$10^0$ cm) locked to the air pressure jump and amplify these long ocean waves by continuously providing them with additional energy (Monserrat et al., 2006). As a result, significant long ocean waves are generated by the Proudman resonance (Proudman, 1929; Vilibić, 2008). Through coupled-mode propagation, the enhanced air pressure jump (Fig. 8d) delivers amplified long ocean waves to multiple harbors. Finally, as sequential harbor meteotsunamis occur along the eastern

Yellow Sea coast, multiple standing oscillations (~$10^2$ cm) are amplified in unique range of the resonant periods and recorded at the harbor tide gauges (Rabinovich, 2020). The meteotsunami warning system in the eastern Yellow Sea provides early warnings, via the short messaging service (SMS), for potential meteotsunamis by detecting the intensity and propagation of air pressure jumps at offshore islands and along the coastline (Kim et al., 2021). According to the current SMS protocol (Fig. 11c), SMS #1, which is for a preliminary caution from the outer caution zone (red crosses in Fig. 1), and SMS #2, which is for a propagation warning from the warning zone along the coastline (green circles in Fig. 1), are based on a dichotomous decision (occurrence/non-occurrence) without a risk level. We proposed an improved meteotsunami warning system that includes risk level by considering the occurrence characteristics (speed and direction) of air pressure jumps and the harbor meteotsunami-dominated environment in the eastern Yellow Sea. The occurrence rate of an air pressure jump cannot be used to monitor a meteotsunami event in real-time because it can only be detected after it has occurred. In the improved meteotsunami warning system, the harbor seiches warning (SMS #3), which has a "very high" risk level, is sent to the harbors located on the propagation path of the pressure jump line when the resonant response (a wave height greater than four sigma with dominant period bands of less than 30 min) is detected at a harbor tide gauge. The harbor tide gauges for the additional warning SMS (SMS #3) were determined based on the spatial pattern of meteotsunami occurrences at each latitude band and the highest Q-factor (Fig. 6b).

This study provides guidance on when, where, and how often meteotsunamis occurred in the eastern Yellow Sea. In addition, a meteotsunami warning system was discussed based on the occurrence characteristics of pressure-forced meteotsunamis. There is a need to confirm the adequacy of the proposed warning at harbor tide gauges (SMS #3) for a timely and reliable meteotsunami warning, because the warnings will often be provided with a limited lead time. This is because most of the harbor tide gauges are located near the coastline, except for the DH harbor tide gauge, which is located the furthest away. Additionally, it is not possible to send a timely warning SMS to the harbor where the harbor meteotsunami is first detected under the current observation system. Nevertheless, the harbor seiche warning is essential for forecasting unexpected and destructive harbor meteotsunamis along the eastern Yellow Sea coast.

*Data availability.* The historical pressure and sea level data used in the current study were obtained from the Korea Meteorological Administration (KMA) and the Korea Hydrographic and Oceanographic Agency (KHOA), respectively. Data are available by request to the institutions.

*Author contributions.* All authors contributed to the conception and design of the study, data acquisition, and manuscript preparation.

*Competing interests.* The authors declare that they have no conflicts of interest.

*Acknowledgement.* The co-authors would like to express their thanks to three referees for their careful reading and valuable comments. This article can be substantially improved compared to the initial submission. Also, thanks to Hee Jung Kim for supporting them as administrative staff.

*Financial support.* This research was part of the project titled "Improvements of ocean prediction accuracy using numerical modeling and artificial intelligence technology," funded by the Ministry of Oceans and Fisheries, Korea. In addition, this
research was supported by a National Research Foundation of Korea Grant from the Korean Government (MSIT ; the Ministry of Science and ICT) NRF-2021M1A5A1075516) (KOPRI-PN21013).

*Review statement.* This paper was edited by Maria Ana Baptista and reviewed by three anonymous referees.

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
