# Peer review of "Occurrence of pressure-forced meteotsunami events in the eastern Yellow Sea during 2010–2019"

_Natural Hazards and Earth System Sciences, 2021_

## Author Comment (AC1)

**Response to Comments**

**Manuscript number: NHESS-2021-126**

**Title:** Pressure-forced meteotsunami occurrences in the eastern Yellow Sea over the past decade (2010–2019): monitoring guidelines **Authors:** Myung-Seok Kim, Seung-Buhm Woo, Hyunmin Eom, and Sung Hyup You **Journal:** Natural Hazards and Earth System Sciences

**- Reviewer #1:**

This paper studied meteotsunamis in the eastern Yellow Sea, and proposed monitoring guidelines in this area. It is well-structured and the results are presented clearly. But it needs a major revision to be considered as a publication in NHESS journal. The authors need to include the analysis on the period of detected waves and the local resonance at the tidal gauges. Authors have written many sentences in a passive voice, and their claims and explanations sound weak.

 $\rightarrow$  We really appreciate your detailed review and comment. As you commented, the analysis on the period of detected waves and local resonance at the tide gauges was performed. More detailed results will be discussed in the revised manuscript.

**[Major comments]**

One of the main characteristics of tsunami waves (including meteotsunamis) is the period of waves since the energy of a tsunami is due to its long period. This study only considered the maximum amplitude waves and did not analyze the period of the waves. The authors need to perform wavelet analysis or Fourier spectrum analysis, and consider peak-to-trough heights rather than maximum amplitudes to confirm meteotsunami cases.

Another important characteristic of meteotsunamis is the local amplification. The local factor can be decisive to forecast the severity of meteotsunamis in the eastern Yellow Sea since the coastline is long and complicated with many islands. The authors can improve this work if they include local factors.

→ Based on wavelet analysis and visual inspection with the meteotsunami events, we examined the dominant periods when the maximum wave heights were detected. As you commented, local amplification is known as an important characteristics of meteotsunamis in the eastern Yellow Sea (Kim et al., 2016, 2021b). Spread of the dominant periods and a quality factor (Q-factor), which is a linear measure of the energy dumping in a basin, will be examined to include local factors. Please also check "Response to Comments (figures, tables, and equations)".

- Kim, M.-S., Kim, H., Kim, Y.-K., Gu, B.-H., Lee, H.-J., Woo, S.-B., 2016. Double resonance effect at Daeheuksando port caused by air pressure disturbances in Yellow Sea on 31 March 2007. J. Coast. Res. 75, 1142–1146. https://doi.org/10.2112/SI75-229.1

- Kim, M.-S., Woo, S.-B., 2021b. Propagation and amplification of meteotsunamis in

multiple harbors along the eastern Yellow Sea coast. Continent. Shelf Res. https://doi.org/10.1016/j.csr.2021.104474

1. The authors studied the local behaviors of tidal gauges (shown in Figure 3), but chose the threshold of 15 cm for all the tidal gauges. Montserrat (2006) suggested 4-sigma and Dusek et al. (2019) suggested 6-sigma and 20 cm (peak-to-trough height) for choosing possible meteotsunami events. Please explain why the authors have chosen the 15 cm threshold.

→ We classified the meteotsunami events by using the maximum amplitude threshold (15 cm) just for the consistency of the threshold used in previous studies in the eastern Yellow Sea. However, we accepted your comments when classifying the meteotsunami events. As you commented, the classification was re-performed using the peak-to-trough wave heights and alternative threshold (20 cm & 4 sigma). The wave height threshold was selected through prototyping with the known meteotsunami events since 2010. As a result, 42 meteotsunami events, which were increased than the previous results (32 events), were classified. Please check the modified figures and tables.

2. In Table 3 and Figure 11, the authors presented average amplitude and occurrence rate to evaluate meteotsunami events. Damages on the coast can occur in a small area, and the occurrence rate can be small. Can these parameters represent the severity of meteotsunamis?

 $\rightarrow$  In this study, we classified 11 extreme events among 42 pressure-forced meteotsunami events based on the occurrence rate (i.e., spatial scale). The average amplitude was not considered. As a result, the occurrence rate of meteotsunamis was related to the occurrence rate of air pressure jump (modified Figure 11). As you commented, damages on the coast can occur in a small area, and the occurrence rate can be small. However, we considered that meteotsunamis that spread over the large area were more dangerous on the eastern Yellow Sea coast. During the pilot operation of the monitoring system in the Yellow Sea, when the long ocean waves amplified by the Proudman resonance propagated with a wider spatial scale, they were more hazardous than the meteotsunamis with local scale (Kim et al., 2021a). As you know, the eastern Yellow Sea coast is characterized by many harbors along the long and complicated coastline. The long ocean waves forced by the propagating air pressure jumps can generate destructive harbor meteotsunamis, causing local amplification in multiple harbors (Kim et al., 2021b). Therefore, the occurrence rate of air pressure jumps can be considered as one of the parameters representing the severity of meteotsunamis from the perspective of monitoring system operation on the eastern Yellow Sea coast.

- Real-time pressure disturbance monitoring system in the Yellow Sea - Pilot test during the period of March to April 2018 (Nat. Hazards SI,2021)

- Propagation and amplification of meteotsunamis in multiple harbors along the eastern Yellow Sea coast (CSR RI,2021)

3. In Table 4, authors proposed guidelines for meteotsunami monitoring. It is unclear why authors choose 30 % occurrence rate for extreme. The occurrence rate cannot be used to forecast events since the occurrence of meteotsunami can be detected after it has

**occured.**

→ As you commented, the occurrence rate cannot be used to forecast events. The warning level will be divided into three levels (high-moderate-low) by using the speed and direction of the pressure disturbances. Instead, for extreme warning levels, we will choose peak-to-trough wave height at beacon tide gauges in which are outer-located harbors (AH, EC, WD, DH, and MS) to consider the local resonance (i.e., multiple harbor resonances). More detailed results will be discussed in the schematic diagram on how the meteotsunami warning system will be designed (Reviewer #2 suggested).

**[Minor comments]**

- 1. L 14 unclear "It appears that the specific characteristics (intensity, occurrence rate, and propagation) of the pressure disturbance are in common on extreme meteotsunami events that are classified by applying the hazardous meteotsunami conditions among the 34 events."
- 2. L 25 "that dominant" -> that are dominant
- 3. L 25-26 remove "which are"
- 4. L 28 remove "as the first stage"
- 5. L 34 remove "worldwide until recently"
- 6. L 35 remove "most"
- 7. L 36 "The meteotsunami event on March 31, 2007, was an event in which" -> On March 31st, 2007,
- 8. L 40 "It was the event that occurred with the strongest intensity in the largest area of the meteotsunami events reported in the Yellow Sea so far" -> It is the strongest meteotsunami event reported in the Yellow Sea so far
- 9. L 43 "This event suggests that the timing of meteotsunami occurrence is an important factor that can determine the level of human casualties." This argument is vague, and the authors need to specify their assertion.
- 10.L 50 remove "Overall"
- 11.L 52 remove "besides the accident events"
- 12.L113 "calculation and threshold" -> calculating the threshold
- 13.L114 "which known"->which is known
- 14.L126 remove "was the meteotsunami event of accident since 2010, which"
- 15.L130-131 remove "In general... and"
- 16.L149 "We need to check .. as a meteotsunami" It is not clear why we need to find it.
- 17.L 229-231 Two sentences are inconsistent. Authors explain the occurrence tendency, then claim that they are irregular. I think 10 years are too short to propose any tendency.
- 18.L 314 "pattern, for example,"->pattern. For example,
- 19.L 356 "specific year" -> "specific season"
- 20.L 390-393 "Another pressure jump ... the west of Lat. A-C" What is the reference for the Greenspan resonance in this area?

 $\rightarrow$  The comments about rephrasing are planned to be modified after the final response.

**Response to Comments (figures, tables, and equations)**

We have significantly revised our results according to your advice. The major revision can be summarized as follows:

(1) Classification of the meteotsunami events

Maximum amplitude  $\rightarrow$  maximum peak-to-trough height Absolute threshold  $\rightarrow$  combined threshold criterion based on four-sigma value and absolute wave height of 20 cm

(2) Meteotsunami occurrences

Yearly and monthly strength parameter of the meteotsunamis were added using the box-plots. Spatial pattern of the meteotsunamis were estimated based on the number of events per year (2d histogram).

(3) Classification of the extreme meteotsunami events considering only occurrence rate

Average amplitude, meteotsunami occurred-tide gauges of more than six, and occurrence rate of more than  $50\% \rightarrow$  meteotsunami occurred-tide gauges of more than six and occurrence rate of more than 50%

(4) Propagation patterns of air pressure jumps on the meteotsuanmi events

Radar image analysis through visual inspection + linear speed and direction using 2 AWS  $\rightarrow$  radar image analysis through visual inspection + pressure tendency method using 3 AWS (Šepić et al., 2009)

(5) Local amplifications in multiple harbors

Scatter diagram of dominant period of detected waves and maximum wave height was added.

The following figures and tables will be significantly polished to improve the readability after the final response.

[Figure 2: The 3rd and 4th panel in following figure will be added in prior Figure 2] Wave height and dominant period of the meteotsunamis during 26/04/2011 meteotsunami event at the DH (upper) and EC (lower) tide gauge. The 1st panel: peak-to-trough height of the filtered sea level in time domain. The 2nd panel: approximated wave period in the time domain. The 3rd panel: wavelet analysis. The 4th panel: distribution of wavelet power spectrum when the maximum wave height observed.

---

## Author Comment (AC2)

**Response to Comments**

**Manuscript number:** NHESS-2021-126
**Title:** Pressure-forced meteotsunami occurrences in the eastern Yellow Sea over the past decade (2010–2019): monitoring guidelines
**Authors:** Myung-Seok Kim, Seung-Buhm Woo, Hyunmin Eom, and Sung Hyup You
**Journal:** Natural Hazards and Earth System Sciences

**- Reviewer #2:**

The manuscript "Pressure-forced meteotsunami occurrences in the eastern Yellow Sea over the past decade (2010-2019): monitoring guidelines" by Kim et al. represent a worthy addition to the meteotsunami research of the eastern Yellow Sea, and I conditionally suggest it for publication. My main concern is the quality of the English language which is rather poor. The manuscript MUST be proofread by either a native speaker with knowledge on the subject or someone with much better working knowledge of the language. I will not list any mistakes, but there are some in almost every sentence. I now list some specific comments:

→ As you commented, we will use one more round of English proofing when submitting the revised manuscript. Please also check "Response to Comments (figures, tables, and equations)". Thank you for your comments.

**[Abstract]**

1. change "which shows a strong seasonal trend." to "revealing a distinct seasonal pattern."
2. list "favorable conditions" which you have found
3. change "the monitoring system" to "the meteotsunami monitoring system"

→ These comments about rephrasing are planned to be modified after the final response.

**[1. Introduction]**

4. change "forced long waves" to "forced ocean long waves"
5. change "to the pressure disturbance" to "to the atmospheric pressure disturbance"
6. change "waves and their fundamental periods" to "waves and fundamental periods of shelves, bays or harbors".
7. change "at that time remains unknown" to "was unknown at that time

→ These comments about rephrasing are planned to be modified after the final response.

**[2. Observation system and pressure jump]**

8. Change the title to "Observation system and extraction of meteotsunami generating pressure disturbances" or simply to "Meteotsunami monitoring system"

→ This comment is planned to be modified after the final response.

9. It is implied (around line 115) that various intensities were tested, but no information on the results of these tests is given. Please explain how did you choose the 1.5 hPa/10 min rate. Also, have you tested intensities over shorter time intervals, e.g. XY hPa/5 min? Please discuss.

→ Kim et al. (2021a) explained how they choose the reason of the 10 min rate (Please refer to 2.3 Preliminary caution SMS). When operating the real-time pressure disturbance monitoring system, it was necessary to consider the delayed time for raw pressure data observed at each AWS to be sent to the KMA (Korea Meteorological Administration). The criterion of air pressure jump in the Yellow Sea was based on the observed intensity of air pressure disturbance during the meteotsunamis of accident (Kim et al., 2019). Also, we tested intensities over shorter and longer time intervals (e.g., hPa/5 min & hPa/20 min) as following figure. We applied the same criterion (red line) of air pressure jump (0.15 hPa/min). Following figure indicates raw pressure data and air pressure disturbances for each time interval at the DH harbor where the largest meteotsunami was detected since 2010 (meteotsunami of accident: 26/04/2011). The shorter the interval of the rate, the more sensitive it is, but from the point of view of real-time monitoring system operation, it was decided to be 10 min rate.

[Figure]

**Figure:** Raw pressure data and air pressure disturbances for each time interval (5, 10, and 20 min rate of pressure change) at the DH AWS during 25-27 April 2011.

- Kim, M.-S., Kim, H., Eom, H.-M., Yoo, S.-H., Woo, S.-B., 2019. Occurrence of

hazardous meteotsunamis coupled with pressure disturbance traveling in the Yellow Sea, Korea. J. Coast. Res. 91, 71–75. https://doi.org/10.2112/si91-015.1

- Kim, M.-S., Eom, H., You, S.-H., Woo, S.-B., 2021a. Real-time pressure disturbance monitoring system in the Yellow Sea: pilot test during the period of March to April 2018. Nat. Hazards. https://doi.org/10.1007/s11069-020-04245-9

**[3. Classification of pressure-forced meteotsunami dates]**

10. change the title; "dates" were not pressure-forced; sea levels were pressure forced or meteotsunami; perhaps: "Classification of pressure-forced" meteotsunami events

    → This comment is planned to be modified after the final response.

11. "which means the inverted barometer response" - no, the inverted barometer response is ~1cm/hPa - what you have here is much stronger - so, this means "resonant effect between the propagating air pressure disturbance and long ocean waves"!

    → This comment about rephrasing is planned to be modified after the final response.

12. "phase relationship between pressure jump and high-frequency sea level.." - what "phase relationship"? "In-phase, out-of-phase, almost simultaneous appearance"?

    → We intended similar timing of occurrence. We will rephase the expression of "phase relationship" to "timing".

13. What kind of filter did you use? Please state and give an appropriate reference.

    → We used the wavelet filter that Torrence and Compo (1998) suggested. This filter has the advantage over traditional filtering in that it can be used to isolate single events that have a broad power spectrum or multiple events that have varying frequency.

    - Torrence, C., Compo, G.P., 1998. A practical guide to wavelet analysis. Bull. Am. Meteorol. Soc. 79, 61–78. https://doi.org/10.1175/1520-0477(1998)079<0061:APGTWA>2.0.CO;2

14. Figure 3. and accompanied analysis/text - I like this idea for extracting the extremes. However, since the events are almost symmetric around zero I would consider looking at the wave heights instead of at amplitudes and extracting the events in a similar way.

    → As you commented, the meteotsunami events were re-analyzed using wave height rather than amplitude. Please check the modified figures and tables.

15. Figure 4. It is not clear from this Figure what was excluded "Sample data collection: 68% (1 sigma)." I suggest writing "Exclusion of dates with less than 68% of available data" - sigma is a strange variable to use when it comes to a number of data.

→ This comment is planned to be modified after the final response.

16. change "was controlled in the first" to "was removed in the first"

→ This comment is planned to be modified after the final response.

17. You say that you removed daily mean from daily samples. That is not really necessary if you filtered the data as well, and you have, as I understand?

→ As you commented, demean from each daily sample is not necessary. The process flow diagram (Figure 4) will be modified after the final response.

**[4. Pressure-forced meteotsunami occurrences]**

18. Table 2. Mark the strongest amplitude of each event with bold letters, or underline. So, in the first row that would be 33.3 at MS station...

→ As you commented, we modified the table. Please check the modified table.

19. Line 234-235. Discuss here or in discussion (better in discussion) why do you think that meteotsunamis are more common during March-May

→ We will discuss the possible reason of the meteotsunami seasonality in discussion. Please refer to 28[th] comment.

20. Figure 6. Think about adding some strength parameter to this Figure - for example, for each month try plotting median height at stations at which it was recorded.

→ As you commented, we modified the figure. Please check the modified figure.

21. Line 240. change "The spatial vulnerability" to "The spatial spread" or "The spatial pattern".

→ This comment is planned to be modified after the final response.

22. Figure 7. Instead of showing a total number of events, show the number of events per year - this way, the effect of shortness of time series will be removed.

→ As you commented, we modified the figure. Please check the modified figure.

23. In your list (line 260) condition (3) is the same as condition (2) but stronger - remove the condition (2).

→ This comment is planned to be modified after the final response.

24. Figure 8. I like the idea

→ Thank you for your encouragement.

25. Figure 9. I suggest adding another column in which filtered air pressure time series are shown, starting at the top with the northern stations, and ending, at the bottom with the southern stations - or another way around.

→ As you commented, we will add the filtered air pressure time series from the starting and ending.

26. It is not clear how were speed and direction of propagation assessed? From radar images or from air pressure data? Please explain. If from the radar data, confirm it with air pressure data.

→ The propagation patterns of the classified 42 meteotsunami events were analyzed as follows:

(1) The intensity and movement of rain rate exceeding 5 mm/h were confirmed by visual inspection (Kim et al., 2021a).

(2) Arrival time list and isochrone map of air pressure jump were estimated in the area where the high rain rate propagated (Figure 9).

(3) Direction and speed were assessed using the three points of AWSs based on the explicit formula suggested by Šepić et al. (2009). Equations are specified in Response to Comments (figures, tables, and equations).

- Kim, M.-S., Eom, H., You, S.-H., Woo, S.-B., 2021a. Real-time pressure disturbance monitoring system in the Yellow Sea: pilot test during the period of March to April 2018. Nat. Hazards. https://doi.org/10.1007/s11069-020-04245-9

- Šepić, J., Denis, L., Vilibić, I., 2009. Real-time procedure for detection of a meteotsunami within an early tsunami warning system. Phys. Chem. Earth 34, 1023–1031. https://doi.org/10.1016/j.pce.2009.08.006

27. Figure 11. I like the idea of this Figure as well.

→ Thank you for your encouragement.

**[5. Discussion]**

28. Please discuss reasons why do you suppose meteotsunamis are most common from March to May.

→ We will discuss possible connection between the following synoptic patterns and meteotsunami occurrences based on previous results (Kim et al., 2016, 2017). According to previous results, the spring season (March to May) in the Korean peninsula has the seasonal characteristics of a migratory anticyclone and an extratropical

depression generated in the Tibet and Mongolian plateau passing through the Yellow Sea every three or four days. The spatial distribution of the atmospheric pressure system generally increases the potential atmospheric instability in the Yellow Sea. Atmospheric instability (e.g., pressure jump, low-level jet), which can lead to fluctuations of the sea level, often increase when a cold front in an extratropical depression passes through the Yellow Sea.

- Kim, H., Kim, M.-S., Lee, H.-J., Woo, S.-B., Kim, Y.-K., 2016. Seasonal characteristics and mechanisms of meteo-tsunamis on the west coast of Korean Peninsula. J. Coast. Res. 75, 1147–1151. https://doi.org/10.2112/SI75-230.1

- Kim, H., Kim, M.-S., Kim, Y.-K., Yoo, S.-H., Lee, H.-J., 2017. Numerical weather prediction for mitigating the fatal loss by the meteo-tsunami incidence on the west coast of Korean Peninsula. J. Coast. Res. 79, 119–123. https://doi.org/10.2112/SI79-025.1

29. Please give a point-by-point schematic (perhaps a figure) on how the meteotsunami warning system will be designed: Thus e.g., constant monitoring of air pressure, an automatic warning to personal when air pressure rate of change surpasses a given threshold at one of the beacon stations, careful examination of all air pressure stations, determination of speed and direction as soon as possible, issuing a warning. As a final note, I compliment the authors for the nice research and figures.

→ We are planning the following meteotsunami warning system. The conceptual diagram will be prepared after the final response.

(1) The existing meteotsunami warning system is operated based on the following characteristics (Kim et al., 2021a):

- Observation system is organized with the 89 AWSs (17 AWSs in caution zone and 72 AWSs in waring zone).
- Air pressure disturbances exceeding 1.5 hPa/10 min are regarded as air pressure jumps that can generate the meteotsunamis in the Yellow Sea.
- The meteotsunami alerts are divided into two levels (preliminary caution SMS and propagation SMS).

(2) It is planned to utilized radar image and outer-located harbor tide gauges (AH, EC, WD, DH, and MS) as additional observation system. The meteotsunami alerts will be divided into three levels (preliminary caution SMS, propagation SMS, and warning SMS):

- The temporal and spatial variability of air pressure jumps in the open sea will be tracked with the radar image.
- Favorable conditions (speed and direction) for the generation of extreme meteotsunami events were examined in this study. When sending the 2nd SMS (propagation SMS), it is planned to include warning level (high-moderate-low) based on the speed and direction of air pressure jumps.
- If the peak-to-trough wave height at one tide gauge among the five tide gauges exceeds the meteotsunami limit (20 cm and 4 sigma) and dominant period bands of the waves are less than 30 min, it is suggested to send a warning SMS (3rd SMS: extreme level). The prior two SMSs are sent through proxy-based assessment. The

last warning SMS will be sent based on resonant waves directly observed at each beacon tide gauge after sending the two SMSs.

- Kim, M.-S., Eom, H., You, S.-H., Woo, S.-B., 2021a. Real-time pressure disturbance monitoring system in the Yellow Sea: pilot test during the period of March to April 2018. Nat. Hazards. https://doi.org/10.1007/s11069-020-04245-9

**Response to Comments (figures, tables, and equations)**

We have significantly revised our results according to your advice. The major revision can be summarized as follows:

(1) Classification of the meteotsunami events

Maximum amplitude → maximum peak-to-trough height
Absolute threshold → combined threshold criterion based on four-sigma value and absolute wave height of 20 cm

(2) Meteotsunami occurrences

Yearly and monthly strength parameter of the meteotsunamis were added using the box-plots. Spatial pattern of the meteotsunamis were estimated based on the number of events per year (2d histogram).

(3) Classification of the extreme meteotsunami events considering only occurrence rate

Average amplitude, meteotsunami occurred-tide gauges of more than six, and occurrence rate of more than 50% → meteotsunami occurred-tide gauges of more than six and occurrence rate of more than 50%

(4) Propagation patterns of air pressure jumps on the meteotsuanmi events

Radar image analysis through visual inspection + linear speed and direction using 2 AWS → radar image analysis through visual inspection + pressure tendency method using 3 AWS (Šepić et al., 2009)

(5) Local amplifications in multiple harbors

Scatter diagram of dominant period of detected waves and maximum wave height was added.

The following figures and tables will be significantly polished to improve the readability after the final response.

[Figure 2: The 3rd and 4th panel in following figure will be added in prior Figure 2] Wave height and dominant period of the meteotsunamis during 26/04/2011 meteotsunami event at the DH (upper) and EC (lower) tide gauge. The 1st panel: peak-to-trough height of the filtered sea level in time domain. The 2nd panel: approximated wave period in the time domain. The 3rd panel: wavelet analysis. The 4th panel: distribution of wavelet power spectrum when the maximum wave height observed.

[Figure]

[Figure]

[Table 2: modified] Daily maximum wave height (cm) during 42 meteotsunami events. The reported events since 2010 are denoted by superscript. The strongest intensity of each event are marked by underlined and bold text. The events are indicated as Day/Month/Year.

[revised manuscript text omitted]

[Figure 6: modified] Temporal meteotsunami occurrences between 2010 and 2019: (a-b) number of events per year and month, (c-d) distribution of wave height according to year and month.

[Figure]

[Figure 7: modified] Spatial meteotsunami occurrences between 2010 and 2019: (a) number of events at each tide gauge per year, (b) total number of events at each tide gauge.

[Figure]

[Table 3: modified] Average intensity and occurrence rate of pressure jump and meteotsunami during extreme meteotsunami events. Extreme meteotsunami event dates are indicated as Day/Month/Year.

| Extreme event date | Pressure jump | | | Meteotsunami | | |
|---|---|---|---|---|---|---|
| | Average intensity (hPa/10 min) | Detected AWSs | Occurrence rate (%) | Average intensity (cm) | Detected tide gauges | Occurrence rate (%) |
| 10/02/2010 | 1.8 | 28/87 | 32 | 36.0 | 6/7 | 86 |
| 11/02/2010 | 2.1 | 28/87 | 32 | 37.9 | 6/8 | 75 |
| 01/03/2010 | 1.7 | 46/86 | 53 | 36.1 | 6/6 | 100 |
| 30/04/2011 | 2.6 | 40/86 | 47 | 30.2 | 6/8 | 75 |
| 03/02/2013 | 2.5 | 29/88 | 33 | 22.7 | 6/12 | 50 |
| 14/04/2013 | 1.7 | 27/88 | 31 | 29.4 | 7/12 | 58 |
| 04/04/2015 | 2.7 | 49/88 | 56 | 26.3 | 8/13 | 62 |
| 12/05/2015 | 1.7 | 12/89 | 13 | 27.4 | 8/12 | 67 |
| 04/03/2018 | 2.6 | 32/89 | 36 | 34.7 | 8/11 | 73 |
| 20/03/2019 | 2.5 | 47/88 | 53 | 28.9 | 7/11 | 64 |
| 10/11/2019 | 2.1 | 34/87 | 39 | 23.8 | 7/13 | 54 |

[Figure 8: modified] Heatmap of extreme meteotsunami events: latitude band-averaged intensity of (a) pressure jump and (b) meteotsunami.

[Equation 1-2: added]

The direction $\theta$ and speed $U$ of air pressure jumps were estimated using a triangle of AWSs with coordinates $(x_1, y_1)$, $(x_2, y_2)$, and $(x_3, y_3)$. Šepić et al. (2009) suggested that the traveling air pressure jump can be tracked based on the assumption that (i) air pressure jump does not change during its travel over the domain, and (ii) air pressure jump has a constant direction and speed. The propagation pattern is expressed as follows:

$$tan\theta = a = \frac{\Delta t_{12}\Delta y_{13} - \Delta t_{13}\Delta y_{12}}{\Delta t_{13}\Delta x_{12} - \Delta t_{12}\Delta x_{13}}, \qquad (1)$$

$$U = \frac{1}{\Delta t_{12}}\frac{\Delta y_{12} - a\Delta x_{12}}{\sqrt{1+a^2}} = \frac{1}{\Delta t_{13}}\frac{\Delta y_{13} - a\Delta x_{13}}{\sqrt{1+a^2}}, \qquad (2)$$

where $\Delta t_{12}$ and $\Delta t_{13}$ are the time lags between each AWS; $\Delta x_{12}$, $\Delta x_{13}$, $\Delta y_{12}$, and $\Delta y_{13}$ are distances between each AWS in the east-west and north-south direction, respectively.

- Šepić, J., Denis, L., Vilibić, I., 2009. Real-time procedure for detection of a meteotsunami within an early tsunami warning system. Phys. Chem. Earth 34, 1023–1031. https://doi.org/10.1016/j.pce.2009.08.006

[Figure 11: modified] Scatter diagram and histograms showing propagation characteristics (speed, direction, and occurrence rate) of air pressure jump on 42 meteotsunami events. Red dashed square encloses dominant range of speed and direction of air pressure jump. Circles mark 11 extreme events classified based on occurrence rate of meteotsunamis. The other 31 events are marked with cross marker. Colors of each marker indicate the occurrence rate of air pressure jumps.

[Figure]

[New Figure] Local amplification of meteotsunamis in semi-closed basins. (a) Scatter diagram of wave period to wave height of the classified 42 meteotsunami events, and histogram. (b) distribution of wave period at each tide gauge.

[Figure]

Other figures and tables will be updated after the final response.

[Figure 1]

[Table 1]

[Figure 3] will be deleted (prior criterion).

[Figure 4] will be modified in the revised manuscript.

[Figure 5] will be modified in the revised manuscript.

[Figure 9] will be modified in the revised manuscript.

[Figure 10] will be modified in the revised manuscript.

[New Figure] indicating the conceptual diagram of the meteotsunami warning system will be added as last figure.

Google Earth satellite images indicating the semi-closed basins in which the tide gauges (red squares) are located will be added as the appendix.

---

## Author Comment (AC4)

**Response to Comments**

**Manuscript number:** NHESS-2021-126
**Title:** Pressure-forced meteotsunami occurrences in the eastern Yellow Sea over the past decade (2010–2019): monitoring guidelines
**Authors:** Myung-Seok Kim, Seung-Buhm Woo, Hyunmin Eom, and Sung Hyup You
**Journal:** Natural Hazards and Earth System Sciences

**- Reviewer #3:**

The manuscripts documents a climatology of meteotsunami events, by a systematic examination of sea level and air pressure data in a 10-years period. I found the presented material interesting, worth of publication, yet - as Reviewer #2 commented - the level of language is really not at satisfactorily level, which does not apply only to pure syntax and grammar, but also on sentence constructions and some terminology. So, the language should be improved before eventual acceptance.
I will not repeat comments of other reviewers, in particular of these being the result of language problems, but to add the following:

→ We want to thank the reviewer for his/her valuable comments and considerable contribution for improving the quality of the research. As you commented, we will use one more round of English proofing when submitting the revised manuscript. Please also check "Response to Comments (figures, tables, and equations)".

**[Specific comments]**

1. "monitoring guidelines" should be omitted for the title, as this is not examined but only discussed in the manuscript.

   → This comment is planned to be modified after the final response.

2. Line 13. "Spatially frequent" cannot be used to describe something happening at a single tide gauge, please rephrase.

   → This comment is planned to be modified after the final response.

3. Line 126 and more. "meteotsunami event of accident" - it should be better to say "destructive meteotsunami events" or else. Even more, for classification of meteotsunami events you may use the newly proposed classification of meteotsunami intensities by Vilibic et al. (NH, 2021, https://doi.org/10.1007/s11069-021-04679-9).

   → The comments about rephrasing are planned to be modified after the final response. We thank the reviewer for providing the literature reference. The intensity scale and spatial coverage scales are useful but need to be adapted to our study area. We plan to apply that scale in our next study.

4. Lines 248-250. Several problems in this sentence, including "yellow sea" with small letters ... Change to something like "... is expected to be a beacon tide gauge.", and

omit "under any pressure disturbances" (as not necessary). What is "first meteotsunami"? (again clumsiness in language)

→ This comment is planned to be modified after the final response.

5. Lines 278-280. That is for sure, and even quantified for the Adriatic - see Fig. 7 in Denamiel et a. (2020, JPO, https://doi.org/10.1175/JPO-D-19-0147.1) .

→ As you commented, we will refer to the publication. Of the six parameters of the atmospheric disturbance (amplitude, direction, speed, period, start location, and width), it seems possible to discuss about five variables except for the start location. Thank you for your advice.

6. Line 325-326. Why? As it is known that it is not key factor in some other parts of the world (see previous comment)

→ [Similar comments from Reviewer #1 (major comment #2)]

→ In this study, we classified 11 extreme events among 42 pressure-forced meteotsunami events based on the occurrence rate (i.e., spatial scale). The average amplitude was not considered. As a result, the occurrence rate of meteotsunamis was related to the occurrence rate of air pressure jump (modified Figure 11). As you commented, damages on the coast can occur in a small area, and the occurrence rate can be small. However, we considered that meteotsunamis that spread over the large area were more dangerous on the eastern Yellow Sea coast. During the pilot operation of the monitoring system in the Yellow Sea, when the long ocean waves amplified by the Proudman resonance propagated with a wider spatial scale, they were more hazardous than the meteotsunamis with local scale (Kim et al., 2021a). As you know, the eastern Yellow Sea coast is characterized by many harbors along the long and complicated coastline. The long ocean waves forced by the propagating air pressure jumps can generate destructive harbor meteotsunamis, causing local amplification in multiple harbors (Kim et al., 2021b). Therefore, the occurrence rate of air pressure jumps can be considered as one of the parameters representing the severity of meteotsunamis from the perspective of monitoring system operation on the eastern Yellow Sea coast.

- Kim, M.-S., Eom, H., You, S.-H., Woo, S.-B., 2021a. Real-time pressure disturbance monitoring system in the Yellow Sea: pilot test during the period of March to April 2018. Nat. Hazards. https://doi.org/10.1007/s11069-020-04245-9

- Kim, M.-S., Woo, S.-B., 2021b. Propagation and amplification of meteotsunamis in multiple harbors along the eastern Yellow Sea coast. Continent. Shelf Res. https://doi.org/10.1016/j.csr.2021.104474

7. Lines 334-347. Is it necessary to provide the explicit formula for computation of air pressure disturbance and speed.

→ The propagation patterns of the classified 42 meteotsunami events were analyzed as follows:

(1) The intensity and movement of rain rate exceeding 5 mm/h were confirmed by visual inspection (Kim et al., 2021a).

(2) Arrival time list and isochrone map of air pressure jump were estimated in the area where the high rain rate propagated (Figure 9).

(3) Direction and speed were assessed using the three points of AWSs based on the explicit formula suggested by Šepić et al. (2009). Equations are specified in Response to Comments (figures, tables, and equations).

- Kim, M.-S., Eom, H., You, S.-H., Woo, S.-B., 2021a. Real-time pressure disturbance monitoring system in the Yellow Sea: pilot test during the period of March to April 2018. Nat. Hazards. https://doi.org/10.1007/s11069-020-04245-9

- Šepić, J., Denis, L., Vilibić, I., 2009. Real-time procedure for detection of a meteotsunami within an early tsunami warning system. Phys. Chem. Earth 34, 1023–1031. https://doi.org/10.1016/j.pce.2009.08.006

8. Lines 363-367. Can you discuss eventual connection between synoptic patterns and meteotsunami occurrence also in Yellow Sea? I.e. by examining climate of synoptic patterns above Yellow Sea or Korean Peninsula (as published in literature) or similar?

→ We will discuss possible connection between the following synoptic patterns and meteotsunami occurrences based on previous results (Kim et al., 2016, 2017). According to previous results, the spring season (March to May) in the Korean peninsula has the seasonal characteristics of a migratory anticyclone and an extratropical depression generated in the Tibet and Mongolian plateau passing through the Yellow Sea every three or four days. The spatial distribution of the atmospheric pressure system generally increases the potential atmospheric instability in the Yellow Sea. Atmospheric instability (e.g., pressure jump, low-level jet), which can lead to fluctuations of the sea level, often increase when a cold front in an extratropical depression passes through the Yellow Sea.

- Kim, H., Kim, M.-S., Lee, H.-J., Woo, S.-B., Kim, Y.-K., 2016. Seasonal characteristics and mechanisms of meteo-tsunamis on the west coast of Korean Peninsula. J. Coast. Res. 75, 1147–1151. https://doi.org/10.2112/SI75-230.1

- Kim, H., Kim, M.-S., Kim, Y.-K., Yoo, S.-H., Lee, H.-J., 2017. Numerical weather prediction for mitigating the fatal loss by the meteo-tsunami incidence on the west coast of Korean Peninsula. J. Coast. Res. 79, 119–123. https://doi.org/10.2112/SI79-025.1

**Response to Comments (figures, tables, and equations)**

We have significantly revised our results according to your advice. The major revision can be summarized as follows:

(1) Classification of the meteotsunami events

Maximum amplitude → maximum peak-to-trough height
Absolute threshold → combined threshold criterion based on four-sigma value and absolute wave height of 20 cm

(2) Meteotsunami occurrences

Yearly and monthly strength parameter of the meteotsunamis were added using the box-plots. Spatial pattern of the meteotsunamis were estimated based on the number of events per year (2d histogram).

(3) Classification of the extreme meteotsunami events considering only occurrence rate

Average amplitude, meteotsunami occurred-tide gauges of more than six, and occurrence rate of more than 50% → meteotsunami occurred-tide gauges of more than six and occurrence rate of more than 50%

(4) Propagation patterns of air pressure jumps on the meteotsuanmi events

Radar image analysis through visual inspection + linear speed and direction using 2 AWS → radar image analysis through visual inspection + pressure tendency method using 3 AWS (Šepić et al., 2009)

(5) Local amplifications in multiple harbors

Scatter diagram of dominant period of detected waves and maximum wave height was added.

The following figures and tables will be significantly polished to improve the readability after the final response.

[Figure 2: The 3ʳᵈ and 4ᵗʰ panel in following figure will be added in prior Figure 2] Wave height and dominant period of the meteotsunamis during 26/04/2011 meteotsunami event at the DH (upper) and EC (lower) tide gauge. The 1ˢᵗ panel: peak-to-trough height of the filtered sea level in time domain. The 2ⁿᵈ panel: approximated wave period in the time domain. The 3ʳᵈ panel: wavelet analysis. The 4ᵗʰ panel: distribution of wavelet power spectrum when the maximum wave height observed.

[Figure]

[Figure]

[Table 2: modified] Daily maximum wave height (cm) during 42 meteotsunami events. The reported events since 2010 are denoted by superscript. The strongest intensity of each event are marked by underlined and bold text. The events are indicated as Day/Month/Year.

[revised manuscript text omitted]

[Figure 6: modified] Temporal meteotsunami occurrences between 2010 and 2019: (a-b) number of events per year and month, (c-d) distribution of wave height according to year and month.

[Figure]

[Figure 7: modified] Spatial meteotsunami occurrences between 2010 and 2019: (a) number of events at each tide gauge per year, (b) total number of events at each tide gauge.

[Figure]

[Table 3: modified] Average intensity and occurrence rate of pressure jump and meteotsunami during extreme meteotsunami events. Extreme meteotsunami event dates are indicated as Day/Month/Year.

| Extreme event date | Pressure jump | | | Meteotsunami | | |
|---|---|---|---|---|---|---|
| | Average intensity (hPa/10 min) | Detected AWSs | Occurrence rate (%) | Average intensity (cm) | Detected tide gauges | Occurrence rate (%) |
| 10/02/2010 | 1.8 | 28/87 | 32 | 36.0 | 6/7 | 86 |
| 11/02/2010 | 2.1 | 28/87 | 32 | 37.9 | 6/8 | 75 |
| 01/03/2010 | 1.7 | 46/86 | 53 | 36.1 | 6/6 | 100 |
| 30/04/2011 | 2.6 | 40/86 | 47 | 30.2 | 6/8 | 75 |
| 03/02/2013 | 2.5 | 29/88 | 33 | 22.7 | 6/12 | 50 |
| 14/04/2013 | 1.7 | 27/88 | 31 | 29.4 | 7/12 | 58 |
| 04/04/2015 | 2.7 | 49/88 | 56 | 26.3 | 8/13 | 62 |
| 12/05/2015 | 1.7 | 12/89 | 13 | 27.4 | 8/12 | 67 |
| 04/03/2018 | 2.6 | 32/89 | 36 | 34.7 | 8/11 | 73 |
| 20/03/2019 | 2.5 | 47/88 | 53 | 28.9 | 7/11 | 64 |
| 10/11/2019 | 2.1 | 34/87 | 39 | 23.8 | 7/13 | 54 |

[Figure 8: modified] Heatmap of extreme meteotsunami events: latitude band-averaged intensity of (a) pressure jump and (b) meteotsunami.

[Figure]

[Equation 1-2: added]

The direction $\theta$ and speed $U$ of air pressure jumps were estimated using a triangle of AWSs with coordinates (x₁,y₁), (x₂,y₂), and (x₃,y₃). Šepić et al. (2009) suggested that the traveling air pressure jump can be tracked based on the assumption that (i) air pressure jump does not change during its travel over the domain, and (ii) air pressure jump has a constant direction and speed. The propagation pattern is expressed as follows:

$$tan\theta = a = \frac{\Delta t_{12}\Delta y_{13} - \Delta t_{13}\Delta y_{12}}{\Delta t_{13}\Delta x_{12} - \Delta t_{12}\Delta x_{13}}, \tag{1}$$

$$U = \frac{1}{\Delta t_{12}}\frac{\Delta y_{12} - a\Delta x_{12}}{\sqrt{1 + a^2}} = \frac{1}{\Delta t_{13}}\frac{\Delta y_{13} - a\Delta x_{13}}{\sqrt{1 + a^2}}, \tag{2}$$

where $\Delta t_{12}$ and $\Delta t_{13}$ are the time lags between each AWS; $\Delta x_{12}$, $\Delta x_{13}$, $\Delta y_{12}$, and $\Delta y_{13}$ are distances between each AWS in the east-west and north-south direction, respectively.

- Šepić, J., Denis, L., Vilibić, I., 2009. Real-time procedure for detection of a meteotsunami within an early tsunami warning system. Phys. Chem. Earth 34, 1023–1031. https://doi.org/10.1016/j.pce.2009.08.006

[Figure 11: modified] Scatter diagram and histograms showing propagation characteristics (speed, direction, and occurrence rate) of air pressure jump on 42 meteotsunami events. Red dashed square encloses dominant range of speed and direction of air pressure jump. Circles mark 11 extreme events classified based on occurrence rate of meteotsunamis. The other 31 events are marked with cross marker. Colors of each marker indicate the occurrence rate of air pressure jumps.

[Figure]

[New Figure] Local amplification of meteotsunamis in semi-closed basins. (a) Scatter diagram of wave period to wave height of the classified 42 meteotsunami events, and histogram. (b) distribution of wave period at each tide gauge.

[Figure]

Other figures and tables will be updated after the final response.

[Figure 1]

[Table 1]

[Figure 3] will be deleted (prior criterion).

[Figure 4] will be modified in the revised manuscript.

[Figure 5] will be modified in the revised manuscript.

[Figure 9] will be modified in the revised manuscript.

[Figure 10] will be modified in the revised manuscript.

[New Figure] indicating the conceptual diagram of the meteotsunami warning system will be added as last figure.

Google Earth satellite images indicating the semi-closed basins in which the tide gauges (red squares) are located will be added as the appendix.

---

## Author Response (AR1)

**Response to Comments**

**Manuscript number:** NHESS-2021-126
**Title:** Occurrence of pressure-forced meteotsunami events in the eastern Yellow Sea during 2010–2019
**Authors:** Myung-Seok Kim, Seung-Buhm Woo, Hyunmin Eom, and Sung Hyup You
**Journal:** Natural Hazards and Earth System Sciences

**- Reviewer #1:**

This paper studied meteotsunamis in the eastern Yellow Sea, and proposed monitoring guidelines in this area. It is well-structured and the results are presented clearly. But it needs a major revision to be considered as a publication in NHESS journal. The authors need to include the analysis on the period of detected waves and the local resonance at the tidal gauges. Authors have written many sentences in a passive voice, and their claims and explanations sound weak.

→ We really appreciate your detailed review and comment. As you commented, the analysis on the period of detected waves and local resonance at the tide gauges was performed. Accordingly, we have revised the whole article. Please recheck the revised manuscript. As the reviewers commented, the title was changed from "Pressure-forced meteotsunami occurrences in the eastern Yellow Sea over the past decade (2010–2019): monitoring guidelines" to "Occurrence of pressure-forced meteotsunami events in the eastern Yellow Sea during 2010–2019".

**[Major comments]**

One of the main characteristics of tsunami waves (including meteotsunamis) is the period of waves since the energy of a tsunami is due to its long period. This study only considered the maximum amplitude waves and did not analyze the period of the waves. The authors need to perform wavelet analysis or Fourier spectrum analysis, and consider peak-to-trough heights rather than maximum amplitudes to confirm meteotsunami cases.
Another important characteristic of meteotsunamis is the local amplification. The local factor can be decisive to forecast the severity of meteotsunamis in the eastern Yellow Sea since the coastline is long and complicated with many islands. The authors can improve this work if they include local factors.

→ Based on wavelet analysis and visual inspection with the meteotsunami events, we examined the dominant periods when the maximum wave heights were detected. As you commented, local amplification is known as an important characteristics of meteotsunamis in the eastern Yellow Sea. Spread of the dominant periods and a quality factor (Q-factor), which is a linear measure of the energy dumping in a basin, will be examined to include local factors. In the revised manuscript, we accepted most of your comments.

(1) Peak-to-trough wave height:

Figures in prior "Response to Comments", Table 1, revised "3.1 Classification and identification of meteotsunami events" section

(2) Wavelet analysis:

Figures in prior "Response to Comments", Fig. 2 (e-f), revised "3.1 Classification

and identification of meteotsunami events" section

(3) Local factor:

      Fig. 6, revised "3.2 Temporal and spatial pattern of meteotsunami occurrences" section

(4) Local amplification:

      Fig. 10, Fig. 11, added "4.3 Local amplification in harbors" section, revised "5 Discussion and conclusions" section

1. The authors studied the local behaviors of tidal gauges (shown in Figure 3), but chose the threshold of 15 cm for all the tidal gauges. Montserrat (2006) suggested 4-sigma and Dusek et al. (2019) suggested 6-sigma and 20 cm (peak-to-trough height) for choosing possible meteotsunami events. Please explain why the authors have chosen the 15 cm threshold.

  → We classified the meteotsunami events by using the maximum amplitude threshold (15 cm) just for the consistency of the threshold used in previous studies in the eastern Yellow Sea. However, we accepted your comments when classifying the meteotsunami events. As you commented, the classification was re-performed using the peak-to-trough wave heights and alternative threshold (20 cm & 4 sigma). The wave height threshold was selected through prototyping with the known meteotsunami events since 2010. As a result, 42 meteotsunami events, which were increased than the previous results (32 events), were classified. Please check the modified results.

2. In Table 3 and Figure 11, the authors presented average amplitude and occurrence rate to evaluate meteotsunami events. Damages on the coast can occur in a small area, and the occurrence rate can be small. Can these parameters represent the severity of meteotsunamis?

  → In this study, we classified 11 extreme (widespread) events (Table 2) among 42 pressure-forced meteotsunami events (Table 1) based on the occurrence rate (i.e., spatial scale) of meteotsunamis (Page 13 Lines 259-278). The average amplitude was not considered. As a result, the occurrence rate of meteotsunamis was related to the occurrence rate of air pressure jump (Fig. 8-9). As you commented, damages on the coast can occur in a small area, and the occurrence rate can be small. However, we considered that meteotsunamis that spread over the large area were more dangerous on the eastern Yellow Sea coast. During the pilot operation of the monitoring system in the Yellow Sea, when the long ocean waves amplified by the Proudman resonance propagated with a wider spatial scale, they were more hazardous than the meteotsunamis with local scale. As you know, the eastern Yellow Sea coast is characterized by many harbors along the long and complicated coastline. The widespread long ocean waves forced by the propagating air pressure jumps can generate destructive harbor meteotsunamis, causing local amplification in multiple harbors.

3. In Table 4, authors proposed guidelines for meteotsunami monitoring. It is unclear why authors choose 30 % occurrence rate for extreme. The occurrence rate cannot be used to forecast events since the occurrence of meteotsunami can be detected after it has occurred.

  → As you commented, the occurrence rate cannot be used to forecast events. The warning level will be divided into three levels ("high"-"moderate"-"low") by using the

speed and direction of air pressure jump on the extreme events (Fig. 9 and Fig. 11). In addition, for additional warning level in harbors ("very high"), we will choose peak-to-trough wave height and its period at beacon tide gauges in which are outer-located harbors (AH, EC, WD, DH, and MS) to consider the local resonance (i.e., multiple harbor resonances). More detailed results are discussed in the schematic diagram on how the meteotsunami warning system is designed, as shown in Fig. 11 (Reviewer #2 suggested).

**[Minor comments]**

1. L 14 unclear "It appears that the specific characteristics (intensity, occurrence rate, and propagation) of the pressure disturbance are in common on extreme meteotsunami events that are classified by applying the hazardous meteotsunami conditions among the 34 events."

   → Please check the revised abstract.

2. L 25 "that dominant" -> that are dominant

   → Page 1 Line 23

3. L 25-26 remove "which are"

   → removed

4. L 28 remove "as the first stage"

   → removed

5. L 34 remove "worldwide until recently"

   → removed

6. L 35 remove "most"

   → removed

7. L 36 "The meteotsunami event on March 31, 2007, was an event in which" -> On March 31st, 2007,

   → Page 2 Line 33

8. L 40 "It was the event that occurred with the strongest intensity in the largest area of the meteotsunami events reported in the Yellow Sea so far" -> It is the strongest meteotsunami event reported in the Yellow Sea so far

   → Page 2 Line 36

9. L 43 "This event suggests that the timing of meteotsunami occurrence is an important factor that can determine the level of human casualties." - This argument is vague, and the authors need to specify their assertion.

   → Page 2 Lines 38-40

10. L 50 remove "Overall"

    → removed

11. L 52 remove "besides the accident events"

→ removed

12. L113 "calculation and threshold" -> calculating the threshold

→ Page 4 Line 105

13. L114 "which known"->which is known

→ Page 4 Lines 106-107

14. L126 remove "was the meteotsunami event of accident since 2010, which"

→ removed

15. L130-131 remove "In general… and"

→ removed

16. L149 "We need to check .. as a meteotsunami" It is not clear why we need to find it.

→ As the methods were changed, Fig. 3 was deleted. Please check the revised paragraph (Page 7 Lines 153-171).

17. L 229-231 Two sentences are inconsistent. Authors explain the occurrence tendency, then claim that they are irregular. I think 10 years are too short to propose any tendency.

→ As you commented, number of events and distribution of wave height per year were deleted (Fig. 5). Please check the revised "3.2 Temporal and spatial pattern of meteotsunami occurrences" section. Page 22 Lines 383-385

18. L 314 "pattern, for example,"->pattern. For example,

→ Please check the revised "4.2 Propagation of the air pressure jump" section.

19. L 356 "specific year" -> "specific season"

→ Please check the revised "5 Discussion and conclusions" section (rephrased).

20. L 390-393 "Another pressure jump … the west of Lat. A-C" What is the reference for the Greenspan resonance in this area?

→ Please check the revised "5 Discussion and conclusions" section (removed).

**Response to Comments**

**Manuscript number:** NHESS-2021-126
**Title:** Occurrence of pressure-forced meteotsunami events in the eastern Yellow Sea during 2010–2019
**Authors:** Myung-Seok Kim, Seung-Buhm Woo, Hyunmin Eom, and Sung Hyup You
**Journal:** Natural Hazards and Earth System Sciences

**- Reviewer #2:**

The manuscript "Pressure-forced meteotsunami occurrences in the eastern Yellow Sea over the past decade (2010-2019): monitoring guidelines" by Kim et al. represent a worthy addition to the meteotsunami research of the eastern Yellow Sea, and I conditionally suggest it for publication. My main concern is the quality of the English language which is rather poor. The manuscript MUST be proofread by either a native speaker with knowledge on the subject or someone with much better working knowledge of the language. I will not list any mistakes, but there are some in almost every sentence. I now list some specific comments:

→ Based on the comments from the reviews, we have revised the whole article. Please recheck the revised manuscript. As the reviewers commented, the title was changed from "Pressure-forced meteotsunami occurrences in the eastern Yellow Sea over the past decade (2010–2019): monitoring guidelines" to "Occurrence of pressure-forced meteotsunami events in the eastern Yellow Sea during 2010–2019". As you commented, we used one more round of English proofing when submitting the revised manuscript. Thank you for your comments and encouragement.

**[Abstract]**

1. change "which shows a strong seasonal trend." to "revealing a distinct seasonal pattern."
2. list "favorable conditions" which you have found
3. change "the monitoring system" to "the meteotsunami monitoring system"

→ Please check the revised abstract.

**[1. Introduction]**

4. change "forced long waves" to "forced ocean long waves"

→ Page 1 Lines 25-31 (rephrased)

5. change "to the pressure disturbance" to "to the atmospheric pressure disturbance"

→ Page 1 Lines 25-31 (rephrased)

6. change "waves and their fundamental periods" to "waves and fundamental periods of shelves, bays or harbors".

→ Page 1 Line 28

7. change "at that time remains unknown" to "was unknown at that time

→ Page 2 Line 44

**[2. Observation system and pressure jump]**

8. Change the title to "Observation system and extraction of meteotsunami generating pressure disturbances" or simply to "Meteotsunami monitoring system"

   → Page 3

9. It is implied (around line 115) that various intensities were tested, but no information on the results of these tests is given. Please explain how did you choose the 1.5 hPa/10 min rate. Also, have you tested intensities over shorter time intervals, e.g. XY hPa/5 min? Please discuss.

   → Pages 4-5 Lines 104-118

   → Kim et al. (2021) explained how they choose the reason of the 10 min rate (Please refer to 2.3 Preliminary caution SMS). When operating the real-time pressure disturbance monitoring system, it was necessary to consider the delayed time for raw pressure data observed at each AWS to be sent to the KMA (Korea Meteorological Administration). The criterion of air pressure jump in the Yellow Sea was based on the observed intensity of air pressure disturbance during the meteotsunamis of accident (Kim et al., 2019). Also, we tested intensities over shorter and longer time intervals (e.g., hPa/5 min & hPa/20 min) as following figure. We applied the same criterion (red line) of air pressure jump (0.15 hPa/min). Following figure indicates raw pressure data and air pressure disturbances for each time interval at the DH harbor where the largest meteotsunami was detected since 2010 (meteotsunami of accident: 26/04/2011). The shorter the interval of the rate, the more sensitive it is, but from the point of view of real-time monitoring system operation, it was decided to be 10 min rate.

[Figure]

Figure: Raw pressure data and air pressure disturbances for each time interval (5, 10, and 20 min rate of pressure change) at the DH AWS during 25-27 April 2011.

- Kim, M.-S., Kim, H., Eom, H.-M., Yoo, S.-H., Woo, S.-B., 2019. Occurrence of hazardous meteotsunamis coupled with pressure disturbance traveling in the Yellow Sea, Korea. J. Coast. Res. 91, 71–75. https://doi.org/10.2112/si91-015.1

- Kim, M.-S., Eom, H., You, S.-H., Woo, S.-B., 2021. Real-time pressure disturbance monitoring system in the Yellow Sea: pilot test during the period of March to April 2018. Nat. Hazards. https://doi.org/10.1007/s11069-020-04245-9

**[3. Classification of pressure-forced meteotsunami dates]**

10. change the title; "dates" were not pressure-forced; sea levels were pressure forced or meteotsunami; perhaps: "Classification of pressure-forced" meteotsunami events

→ Page 5

11. "which means the inverted barometer response" - no, the inverted barometer response is ~1cm/hPa - what you have here is much stronger - so, this means "resonant effect between the propagating air pressure disturbance and long ocean waves"!

→ Page 5 Lines 136-137

12. "phase relationship between pressure jump and high-frequency sea level.." - what "phase relationship"? "In-phase, out-of-phase, almost simultaneous appearance"?

→ Page 6 Line 140

13. What kind of filter did you use? Please state and give an appropriate reference.

→ Page 5 Lines 131-132, Page 9 Lines 189-192

14. Figure 3. and accompanied analysis/text - I like this idea for extracting the extremes. However, since the events are almost symmetric around zero I would consider looking at the wave heights instead of at amplitudes and extracting the events in a similar way.

→ As you commented, the meteotsunami events were re-analyzed using wave height rather than amplitude. Please check the modified figures and tables.

15. Figure 4. It is not clear from this Figure what was excluded "Sample data collection: 68% (1 sigma)." I suggest writing "Exclusion of dates with less than 68% of available data" - sigma is a strange variable to use when it comes to a number of data.

→ Page 8 modified Fig. 3

16. change "was controlled in the first" to "was removed in the first"

→ Page 9 Lines 188-189

17. You say that you removed daily mean from daily samples. That is not really necessary if you filtered the data as well, and you have, as I understand?

→ Page 8 modified Fig. 3

**[4. Pressure-forced meteotsunami occurrences]**

18. Table 2. Mark the strongest amplitude of each event with bold letters, or underline. So, in the first row that would be 33.3 at MS station...

→ As you commented, we modified the table. Please check the modified Table. 1.

19. Line 234-235. Discuss here or in discussion (better in discussion) why do you think that meteotsunamis are more common during March-May

→ We will discuss the possible reason of the meteotsunami seasonality in discussion. Please refer to 28[th] comment.

20. Figure 6. Think about adding some strength parameter to this Figure - for example, for each month try plotting median height at stations at which it was recorded.

→ As you commented, we modified the figure. Page 12 modified Fig. 5

21. Line 240. change "The spatial vulnerability" to "The spatial spread" or "The spatial pattern".

→ Page 12 Line 230

22. Figure 7. Instead of showing a total number of events, show the number of events per year - this way, the effect of shortness of time series will be removed.

→ As you commented, we modified the figure. Page 13 modified Fig. 6

23. In your list (line 260) condition (3) is the same as condition (2) but stronger - remove the condition (2).

→ Page 14 Lines 260-263

In revised manuscript, the extreme (widespread) meteotsunami events were classified by using the condition (2) and (3). The condition (2) is essential when there are little tide gauges available on the event date (e.g., absolute threshold criterion). Please check the modified Table 1. Dash mark in the table indicates a date with less than 68% of available daily data at each tide gauge.

24. Figure 8. I like the idea

→ Thank you for your encouragement.

25. Figure 9. I suggest adding another column in which filtered air pressure time series are shown, starting at the top with the northern stations, and ending, at the bottom with the southern stations - or another way around.

→ Page 17 modified Fig. 8

26. It is not clear how were speed and direction of propagation assessed? From radar images or from air pressure data? Please explain. If from the radar data, confirm it with air pressure data.

→ The propagation patterns of the classified 42 meteotsunami events were analyzed as follows:

(1) The intensity and movement of rain rate exceeding 5 mm/h were confirmed by visual inspection (Kim et al., 2021).

(2) Arrival time list and isochrone map of air pressure jump were estimated in the area where the high rain rate propagated (Figure 8).

(3) Direction and speed were assessed using the three points of AWSs based on the explicit formula suggested by Šepić et al. (2009). Equations are specified in Page 18.

- Kim, M.-S., Eom, H., You, S.-H., Woo, S.-B., 2021. Real-time pressure disturbance monitoring system in the Yellow Sea: pilot test during the period of March to April 2018. Nat. Hazards. https://doi.org/10.1007/s11069-020-04245-9

- Šepić, J., Denis, L., Vilibić, I., 2009. Real-time procedure for detection of a meteotsunami within an early tsunami warning system. Phys. Chem. Earth 34, 1023–1031. https://doi.org/10.1016/j.pce.2009.08.006

27. Figure 11. I like the idea of this Figure as well.

→ Thank you for your encouragement.

**[5. Discussion]**

28. Please discuss reasons why do you suppose meteotsunamis are most common from March to May.

→ Page 22 Lines 383-398

29. Please give a point-by-point schematic (perhaps a figure) on how the meteotsunami warning system will be designed: Thus e.g., constant monitoring of air pressure, an automatic warning to personal when air pressure rate of change surpasses a given threshold at one of the beacon stations, careful examination of all air pressure stations, determination of speed and direction as soon as possible, issuing a warning. As a final note, I compliment the authors for the nice research and figures.

→ Page 24 added Fig. 11

→ Pages 24-25 Lines 425-453

**Response to Comments**

**Manuscript number:** NHESS-2021-126
**Title:** Occurrence of pressure-forced meteotsunami events in the eastern Yellow Sea during 2010–2019
**Authors:** Myung-Seok Kim, Seung-Buhm Woo, Hyunmin Eom, and Sung Hyup You
**Journal:** Natural Hazards and Earth System Sciences

**- Reviewer #3:**

The manuscripts documents a climatology of meteotsunami events, by a systematic examination of sea level and air pressure data in a 10-years period. I found the presented material interesting, worth of publication, yet - as Reviewer #2 commented - the level of language is really not at satisfactorily level, which does not apply only to pure syntax and grammar, but also on sentence constructions and some terminology. So, the language should be improved before eventual acceptance.
I will not repeat comments of other reviewers, in particular of these being the result of language problems, but to add the following:

→ We want to thank the reviewer for your valuable comments and considerable contribution for improving the quality of the research. As you commented, we used one more round of English proofing when submitting the revised manuscript.

**[Specific comments]**

1. "monitoring guidelines" should be omitted for the title, as this is not examined but only discussed in the manuscript.

→ As the reviewers commented, the title was changed from "Pressure-forced meteotsunami occurrences in the eastern Yellow Sea over the past decade (2010–2019): monitoring guidelines" to "Occurrence of pressure-forced meteotsunami events in the eastern Yellow Sea during 2010–2019".

2. Line 13. "Spatially frequent" cannot be used to describe something happening at a single tide gauge, please rephrase.

→ Please check the revised abstract.

3. Line 126 and more. "meteotsunami event of accident" - it should be better to say "destructive meteotsunami events" or else. Even more, for classification of meteotsunami events you may use the newly proposed classification of meteotsunami intensities by Vilibic et al. (NH, 2021, https://doi.org/10.1007/s11069-021-04679-9).

→ The comments about rephrasing are planned to be modified after the final response. We thank the reviewer for providing the literature reference. The intensity scale and spatial coverage scales are useful but need to be adapted to our study area. We plan to apply that scale in our next study.

4. Lines 248-250. Several problems in this sentence, including "yellow sea" with small

letters ... Change to something like "... is expected to be a beacon tide gauge.", and omit "under any pressure disturbances" (as not necessary). What is "first meteotsunami"? (again clumsiness in language)

→ Please check the revised "3.2 Temporal and spatial pattern of meteotsunami occurrences" section (Pages 11-12).

5. Lines 278-280. That is for sure, and even quantified for the Adriatic - see Fig. 7 in Denamiel et a. (2020, JPO, https://doi.org/10.1175/JPO-D-19-0147.1).

→ Page 18 Line 316, Page 20 Lines 349-351

→ As you commented, we checked the publication. Thank you for your advice. Of the six parameters of the atmospheric disturbance (amplitude, direction, speed, period, start location, and width), it seems possible to discuss about five variables except for the start location. Thank you for your advice.

6. Line 325-326. Why? As it is known that it is not key factor in some other parts of the world (see previous comment)

→ Pages 13-14 Lines 249-267

→ In this study, we classified 11 extreme events among 42 pressure-forced meteotsunami events based on the occurrence rate (i.e., spatial scale). The average amplitude was not considered. As a result, the occurrence rate of meteotsunamis was related to the occurrence rate of air pressure jump (Figure 8 and Figure 9). As you commented, damages on the coast can occur in a small area, and the occurrence rate can be small. However, we considered that meteotsunamis that spread over the large area were more dangerous on the eastern Yellow Sea coast. During the pilot operation of the monitoring system in the Yellow Sea, when the long ocean waves amplified by the Proudman resonance propagated with a wider spatial scale, they were more hazardous than the meteotsunamis with local scale (Kim et al., 2021). As you know, the eastern Yellow Sea coast is characterized by many harbors along the long and complicated coastline. The long ocean waves forced by the propagating air pressure jumps can generate destructive harbor meteotsunamis, causing local amplification in multiple harbors (Kim and Woo, 2021). Therefore, the occurrence rate of air pressure jumps can be considered as one of the parameters representing the severity of meteotsunamis from the perspective of monitoring system operation on the eastern Yellow Sea coast.

- Kim, M.-S., Eom, H., You, S.-H., Woo, S.-B., 2021. Real-time pressure disturbance monitoring system in the Yellow Sea: pilot test during the period of March to April 2018. Nat. Hazards. https://doi.org/10.1007/s11069-020-04245-9

- Kim, M.-S., Woo, S.-B., 2021. Propagation and amplification of meteotsunamis in multiple harbors along the eastern Yellow Sea coast. Continent. Shelf Res. https://doi.org/10.1016/j.csr.2021.104474

7. Lines 334-347. Is it necessary to provide the explicit formula for computation of air

pressure disturbance and speed.

→ The propagation patterns of the classified 42 meteotsunami events were analyzed as follows:

(1) The intensity and movement of rain rate exceeding 5 mm/h were confirmed by visual inspection (Kim et al., 2021).

(2) Arrival time list and isochrone map of air pressure jump were estimated in the area where the high rain rate propagated (Figure 8).

(3) Direction and speed were assessed using the three points of AWSs based on the explicit formula suggested by Šepić et al. (2009). Equations are specified in Page 18.

- Kim, M.-S., Eom, H., You, S.-H., Woo, S.-B., 2021. Real-time pressure disturbance monitoring system in the Yellow Sea: pilot test during the period of March to April 2018. Nat. Hazards. https://doi.org/10.1007/s11069-020-04245-9

- Šepić, J., Denis, L., Vilibić, I., 2009. Real-time procedure for detection of a meteotsunami within an early tsunami warning system. Phys. Chem. Earth 34, 1023–1031. https://doi.org/10.1016/j.pce.2009.08.006

8. Lines 363-367. Can you discuss eventual connection between synoptic patterns and meteotsunami occurrence also in Yellow Sea? I.e. by examining climate of synoptic patterns above Yellow Sea or Korean Peninsula (as published in literature) or similar?

→ Page 22 Lines 383-398

---

## Author Response (AR2)

**Response to Comments**

**Manuscript number:** NHESS-2021-126
**Title:** Occurrence of pressure-forced meteotsunami events in the eastern Yellow Sea during 2010–2019
**Authors:** Myung-Seok Kim, Seung-Buhm Woo, Hyunmin Eom, and Sung Hyup You
**Journal:** Natural Hazards and Earth System Sciences

**- Reviewer:** Introduction, page 2. When talking about general meteotsunami forecasting and early warning systems, I think that the paper by Denamiel et al (2019, https://doi.org/10.1029/2019JC015574) should be mentioned and described in a sentence, as providing a new avenue for meteotsunami early warning systems.

→ Page 2 Lines 61-63, Page 28 Lines 491-493